# CAN ONE MODALITY MODEL SYNERGIZE TRAINING OF OTHER MODALITY MODELS?

**Jae-Jun Lee**[1], **Sung Whan Yoon**[1,2*]
[1]Graduate School of Artificial Intelligence and [2]Department of Electrical Engineering
Ulsan National Institute of Science and Technology (UNIST)
{johnjaejunlee95, shyoon8}@unist.ac.kr

## ABSTRACT

Learning with multiple modalities has recently demonstrated significant gains in many domains by maximizing the shared information across modalities. However, the current approaches strongly rely on high-quality paired datasets, which allow co-training from the paired labels from different modalities. In this context, we raise a pivotal question: *Can a model with one modality synergize the training of other models with the different modalities, even without the paired multimodal supervision?* Our answer is 'Yes'. As a figurative description, we argue that a writer, i.e., a language model, can promote the training of a painter, i.e., a visual model, even without the paired ground truth of text and image. We theoretically show that a superior representation can be achieved by the synergy between two different modalities, without paired supervision. As proofs of concept, we broadly confirm the considerable gains from the synergy across visual, language, and audio models. From a theoretical viewpoint, we first establish a mathematical foundation of the synergy between two different modality models, where each one is trained with its own modality. From a practical viewpoint, our work aims to broaden the scope of multimodal learning to encompass the synergistic usage of single-modality models, relieving a strong limitation of paired supervision. The code is available at https://github.com/johnjaejunlee95/synergistic-multimodal.

## 1 INTRODUCTION

In recent years, *multimodal learning*, which aims to train the joint information across different modalities, is changing the paradigm of cutting-edge deep models from the conventional one modality training to the joint training of multiple modalities. Based on the recent success of large models, such as residual architectures (He et al., 2016), Transformers (Vaswani et al., 2017; Devlin et al., 2018; Liu et al., 2019) and Vision Transformers (ViT) (Dosovitskiy et al., 2021; Steiner et al., 2022), has spurred the emergence of numerous multimodal learning methods, mainly focusing on the vision-language domain.

Notable practices include CLIP (Radford et al., 2021) and ALBEF (Li et al., 2021), which pursue to align the representations from the vision and language modalities. Also, as another branch, CoOp (Zhou et al., 2022b) and CoCoOp (Zhou et al., 2022a) are based on adjustments of the prompt tokens of pretrained vision-language models. More recently, ViT+LLaMA (Pang et al., 2024) attempt to concatenate the pretrained language models with vision models, hypothesizing that the language models would filter significant information from the features extracted by the vision encoder.

This practical success relies on the presumption that the correlated modalities create a synergy when jointly trained on multimodal data samples, e.g., a synergy of the visual information and the textual description of an image, or a synergy of the RGB-based camera images and the LIDAR-based sensing signals of autonomous vehicles. As a theoretical foundation of the practices, researchers claim the existence of the true latent representations, which are able to encompass multiple modalities (Huang et al., 2021; Huh et al., 2024). This implies that when a paired supervision from multiple modalities is given, a model has the potential to more accurately represent the shared semantics by incorporating additional information from other modality models.

---

*: Corresponding Author

We want to raise a critical limitation of the prior works in both practical and theoretical sides. First, the existing multimodal learning methods mainly require the perfectly paired datasets, leading to the immense efforts in building multimodal datasets of high quality and restricting the usage of descent single-modality models in unleashing the synergy between modalities. Second, the current theory only explains how multiple modalities promote better representations when paired labels across modalities are given.

In this context, we start by raising a pivotal question: *Can one modality model synergize training of other modality model, even without matched supervisions across modalities?* As an informal description of the question: Can a writer without true visual supervision (a language model), promote the training of a painter without true textual supervision, (a visual model)? Our answer is 'Yes'.

To showcase how one modality model promotes the training of the other models, we here provide a preliminary experiment, which is simple yet intuitive. We consider a case that a language model **[L]** synergizes the training of a visual model **[V]**; denoting it as **[L→V]** case. Specifically, a pretrained BERT model is used to promote the training of a ResNet model for the CIFAR-10 classification task (Krizhevsky et al., 2009). We used the text prompt `"This is about Class #."`, providing imper-

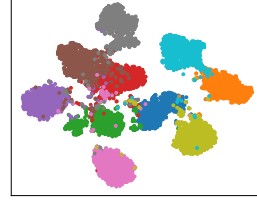 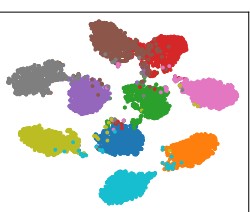

(a) **[V]** (Acc: 96.01)      (b) **[L→V]** (Acc: 97.09)

Figure 1: t-SNE visualizations between single (left) and multimodality (right) on CIFAR-10

fect supervision loosely associated with the image. Herein, # indicates the Arabic class index, which does not provide semantic information of a given image. When utilizing this *imperfect* textual representation in training of the visual model, we surprisingly observe meaningful performance gains. As shown in Figure 1, when compared with the single-modality **[V]** case, our case **[L→V]** shows the well-clustered feature representations and accuracy gain[1]. Noteworthy, it demonstrates that language models help visual models, even with imperfect prompts, i.e., *a writer indeed helps a painter.* Our work aims to provide a theoretical foundation for understanding how it happens, and further proof-of-concepts in variety modalities, architectures, and tasks.

In this paper, we establish a theory that makes us envision how one modality promotes others without paired supervision. The key of this claim is that there exists an interpolated representation of two single-modality representations that outperforms the two (referred to **Theorem 1**). Furthermore, it can be well-approximated without paired supervisions from the given modalities (referred to **Theorem 2**). As the proof of concepts of our claim, we select the most widely-used modalities, i.e., vision **[V]**, language **[L]**, and audio **[A]**, to empirically evaluate the synergies between the three modalities. When using notation, $[M_i \rightarrow M_j]$, indicates that modality $M_i$ promotes the training of modality $M_j$, we mainly confirm that language models promote the training of visual models, i.e., **[L→V]**. Moreover, we find that visual models or language models aid the training of audio models, and vice versa, i.e., **[V→A], [A→V], [L→A]** and **[A→L]**. Noteworthy, our theoretical claim is not limited to particular modalities, which broadens the fundamental understanding of the synergy between different modalities. Also, our work offers the opportunity to utilize the descent single-modality models to enhance other modality models, which strongly relieves the crucial demands of paired supervision of the current multimodal learning.

## 2 RELATED WORKS

### 2.1 VISION LANGUAGE MODEL

Vision and Language are among the most common modalities in deep learning, driving advancements in both empirical and theoretical perspectives. These developments led to foundation models like Transformers for language (Vaswani et al., 2017; Devlin et al., 2018; Liu et al., 2019) and Vision Transformer (ViT) for vision (Dosovitskiy et al., 2021; Steiner et al., 2022), enabling co-training with aligned datasets. A key example is CLIP (Radford et al., 2021), which contrastively learns latent features by maximizing similarity between matching vision-Language pairs while increasing separation for non-matching pairs. Building on this, methods have emerged that incorporate label

---

[1]Full training details are described in Section 4

spaces (Yang et al., 2022) or hard negative samples (Robinson et al., 2021; Li et al., 2021) to improve representation learning. However, these approaches require high-quality, perfectly aligned datasets. Our work overcomes this limitation by showing that even simplified or loosely aligned information can significantly enhance performance.

## 2.2 TRANSLATION OF LANGUAGE INTO VISION MODALITY

Recent advancements in vision-language foundation models have extended their applications across modalities. Large Language Models (LLMs), such as GPT-3 (Brown et al., 2020), and the LLaMA series (Touvron et al., 2023a;b; Dubey et al., 2024), have achieved state-of-the-art results in tasks like reasoning (Talmor et al., 2019) and knowledge retrieval (Kwiatkowski et al., 2019), primarily through prompt engineering. Building on this, several approaches have employed LLMs trained solely on language data to assist in vision tasks (Sharma et al., 2024), using their capabilities to generate new prompts or serve as auxiliary tools. Conversely, our approach focuses on utilizing latent features derived from language modality, even when imperfect data, without relying on the extensive capabilities of LLMs to perform vision-related tasks. This enables a more targeted and efficient use of cross-modal guidance.

## 2.3 MULTIMODAL LEARNING

Several approaches have moved beyond vision-language paradigms to address a wider range of multimodal tasks, both empirically and theoretically. On the practical side, methods like MFAS and Multibench (Liang et al., 2021; Pérez-Rúa et al., 2019) employ fusion networks (Zadeh et al., 2017; Tsai et al., 2019) to combine inputs or latent features from multiple modalities, enhancing multimodal learning. Additionally, related research has explored approaches for handling missing or unpaired sets in multimodal learning (Girdhar et al., 2023; Mizrahi et al., 2024). These methods include training separate classifiers for each modality (Kim & Kim, 2025), training prompts (Lee et al., 2023), or utilizing a shared encoder (Wang et al., 2023) to address missing modalities.

However, prior works also contain a few key limitation. Recent methods in fully supervised or unpaired settings often aim to improve multimodal model performance, typically by training or fine-tuning both modalities (Shukor et al., 2023). In contrast, our approach utilizes latent representations from a well-trained modality model to enhance the training of another modality model from scratch. Moreover, synergy between modality models remains simultaneously underexplored from both empirical and theoretical sight. Thus, our approach addresses this gap by enabling single-modality models to support the training of others empirically, and provide theoretical framework based on interpolated representations to explain their synergy.

## 3 THEORETICAL PERSPECTIVE AND METHODS

### 3.1 BASIC NOTATIONS AND A SKETCH OF MATHEMATICAL CLAIMS

Let us denote $M_i$ and $M_j$ as two different modalities. For inputs, let $x^i$ and $x^j$ indicate the paired inputs from modalties $M_i$ and $M_j$ respectively. In additon, $g_i : \mathcal{X}^i \to \mathcal{Z}^i$ and $g_j : \mathcal{X}^j \to \mathcal{Z}^j$ are two representation models that map an input to latent space, for the respective modalities. Also, $h_i : \mathcal{Z}^i \to \mathcal{Y}$ is the hypothesis from embedding space to the label space $\mathcal{Y}$ for the respective modaltiy. Here, representations and hypothesis are independently trained on each modaltiy, without joint training.

In the context of multimodal learning, it is widely accepted to assume the existence of *true latent space* $\mathcal{Z}^\star$, which is the optimal representation across both modalities $M_i$ and $M_j$ (Huang et al., 2021). As a brief sketch, we show that there exists an interpolated latent representation space $\mathcal{Z}^k$ that shows smaller distance to $\mathcal{Z}^\star$, leading to outperform the two single-modality representations, i.e., $\mathcal{Z}^i$ and $\mathcal{Z}^j$ (referred to **Theorem** 1). It implies that we can find a better representation by interpolating two representations from different modalities. However, the first

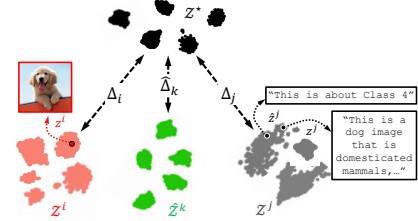

Figure 2: A conceptual sketch of our claims in **[L→V]** case

claim is limited in assuming correctly-paired inputs $x^i$ and $x^j$, which draws extensive costs in annotations of multimodal learning. To this end, we argue that even given with an *imperfect* or *restricted* embedding $\hat{z}^j \in \hat{\mathcal{Z}}^j$ (not correctly-paired with $z^i$), there exists an interpolated representation $\hat{\mathcal{Z}}^k$ between $\mathcal{Z}^i$ and $\hat{\mathcal{Z}}^j$, which is closer to the true representation, leading to surpass the two, i.e., $\mathcal{Z}^i$ and $\mathcal{Z}^j$ (referred to **Theorem** 2). Figure 2 illustrates a brief sketch of our claims. It means that imperfect samples from modality $M_j$ can promote the training of the superior representation that promotes the performance of modality $M_i$; enabling the $[M_j \rightarrow M_i]$ case. In the following part, we provide the formal description.

## 3.2 How One Modality Model Synergize the Training of Others?

We begin by introducing the distance metric between two distributions $\mathcal{P}_1$ and $\mathcal{P}_2$ as follows:

**Definition 1** (2-Wasserstein Distance (Villani et al., 2009)). *2-Wasserstein distance between 2 distribution $\mathcal{P}_1$ and $\mathcal{P}_2$ is defined as:*

$$W_2(\mathcal{P}_1, \mathcal{P}_2) = \inf_{\gamma \in \Gamma(\mathcal{P}_1, \mathcal{P}_2)} \left( \int_{\mathcal{Z}^1 \times \mathcal{Z}^2} d(z_1, z_2)^2 d\gamma(z_1, z_2) \right)^{1/2} \tag{1}$$

*where $\Gamma(\mathcal{P}_1, \mathcal{P}_2)$ is all set of joint distribution (couplings) of $\mathcal{P}_1$ and $\mathcal{P}_2$.*

We use 2-Wasserstein Distance to represent the distance between the probability density functions of the extracted features from different representations, as described below:

**Definition 2.** *Let $z^i \in \mathcal{Z}^i$, $z^j \in \mathcal{Z}^j$, and $z^\star \in \mathcal{Z}^\star$ denote the latent spaces of the i,j-modality, and the true latent space, where $z^\star$ represents the optimal representation across both modalities. $z^i$ and $z^j$ are assumed to be correctly-paired, which means that they represent the same underlying concept but in different modalities (e.g., i for image and j for text). Furthermore, let $\mathcal{P}_i$, $\mathcal{P}_j$, and $\mathcal{P}_\star$ are their corresponding distributions, i.e., $z^i \sim \mathcal{P}_i$, $z^j \sim \mathcal{P}_j$, and $z^\star \sim \mathcal{P}_\star$. The distances between the distributions are then defined as follows:*

$$\Delta_i := W_2(\mathcal{P}_\star, \mathcal{P}_i), \quad \Delta_j := W_2(\mathcal{P}_\star, \mathcal{P}_j) \quad \text{and} \quad \Delta_{ij} := W_2(\mathcal{P}_i, \mathcal{P}_j), \tag{2}$$

*where $\Delta_i$, $\Delta_j$, and $\Delta_{ij}$ are positive real numbers.*

Let us define an interpolated representation between the representations of two different modalities:

**Definition 3.** *Let latent space $z^k \in \mathcal{Z}^k$ is an interpolated representation between the latent representation spaces of the i,j-modalities with interpolation coefficient $\alpha \in [0, 1]$, as follows:*

$$\mathcal{Z}^k := \{z^k = (1 - \alpha)z^i + \alpha z^j \mid z^i \in \mathcal{Z}^i, z^j \in \mathcal{Z}^j\}. \tag{3}$$

In addition, let $\mathcal{P}_k$ denote the distribution of $z^k$, and 2-Wasserstein distance between $\mathcal{P}_k$ and $\mathcal{P}_\star$ is $\Delta_k := W_2(\mathcal{P}_\star, \mathcal{P}_k)$.

**Assumption 1.** *Let $\Delta_{ij}$ not converge to 0, i.e., $\Delta_{ij} \gg 0$.*

Assumption 1 implies that even if modalities $M_i$ and $M_j$ are similarly distant from the true latent space, the distance between $M_i$ and $M_j$ remains significant. Intuitively, while two modalities may represent similar information in certain contexts, their overall representations can differ markedly. Furthermore, they may contain distinct information. For example, modality **[L]** can effectively convey representations about "questions," whereas modality **[V]** may hard to provide such representations. Thus, assuming that $\Delta_{ij} \gg 0$ is a reasonable and justifiable consideration.

In the following theorem, we present the closed form solution of the interpolation coefficient $\alpha^*$, which makes $\mathcal{Z}^k$ be closest to $\mathcal{Z}^\star$.

**Theorem 1.** *The optimal $\alpha^*$ that minimizes $\Delta_k$ is formulated as follows:*

$$\alpha^* = \frac{\Delta_i^2 - \Delta_j^2 + \Delta_{ij}^2}{2\Delta_{ij}^2} \tag{4}$$

*Moreover, the resulting interpolated representation satisfies $\Delta_k \leq \Delta_i$ and $\Delta_k \leq \Delta_j$.*

**Corollary 1.1.** *The optimal $\alpha^*$ is bounded as follows:*

$$\alpha^* = \begin{cases} \left[ \dfrac{\Delta_i}{\Delta_i + \Delta_j}, \, 1 \right] & \textit{if } \Delta_i > \Delta_j \\[4mm] \left[ 0, \, \dfrac{\Delta_i}{\Delta_i + \Delta_j} \right] & \textit{otherwise} \end{cases} \tag{5}$$

For the proofs of Theorem 1 and Corollary 1.1, please see Appendix A.1.

**Remark 1.1.** *(**The behavior of $\alpha^*$**) $\alpha^*$ is strongly influenced by the quality of modalities. For an instance, let us discuss what happens when $\Delta_j$ changes while fixing $\Delta_i$. Specifically, **1)** when $\Delta_i > \Delta_j$, i.e., modaltiy $M_j$ is superior to modality $M_i$, the optimal $\alpha^*$ will be greater than $0.5$, i.e., $\alpha^* > 0.5$, implying that $\mathcal{P}_k$ tends to be closer to $\mathcal{P}_j$. Conversely, **2)** if $\Delta_i < \Delta_j$, $\alpha^* < 0.5$, indicating that $\mathcal{P}_k$ shifts closer to $\mathcal{P}_i$. Finally, **3)** when $\Delta_i = \Delta_j$, the optimal $\alpha^*$ equals to $0.5$. It is convincing that the behavior of $\alpha^*$ shows that the superior interpolated representation forms near to the better modality. Further discussions are provided at Appendix A.1.*

**Remark 1.2.** *(**The condition of the synergy**) For a scenario of utilizing $M_j$ in training $M_i$, it is crucial to judge how beneficial $M_j$ is in promoting the training of the $M_i$. If modality $M_j$ does not bear information on the true latent space, where $\Delta_j \gg \Delta_i$, this suggests that the $M_j$ is much far from both the true latent space and the $M_i$. Thus, leveraging $M_j$ is not effective in training $M_i$, and it makes to $\alpha^*$ close to $0$, which leads to the single-modality training of $M_i$. Thus, Assumption 1 is critical to ensure that information from the $M_j$ can be effectively utilized.*

As aforementioned, obtaining exactly paired datasets in real-world scenarios is challenging. Theorem 1 assumes the perfectly matched pairing of $z_i$ and $z_j$ with a common concept. Let us draws a setting of 'imperfect' or 'restricted' pairing between modalities into our theoretical framework.

**Definition 4.** *Let $\hat{z}^j \in \hat{\mathcal{Z}}^j$ be the imperfect latent space of the $j$-modality, with $\hat{\mathcal{P}}_j$ as its corresponding distribution. The gap $\delta$ between $z^j$ and $\hat{z}^j$ is defined as:*

$$\delta := W_2(\mathcal{P}_j, \hat{\mathcal{P}}_j). \tag{6}$$

*Additionally, the distance from $\hat{\mathcal{P}}_j$ to the true latent space $\mathcal{P}_\star$ and $\mathcal{P}_i$ are denoted as respectively:*

$$\hat{\Delta}_j := W_2(\mathcal{P}_\star, \hat{\mathcal{P}}_j) \quad \textit{and} \quad \hat{\Delta}_{ij} := W_2(\mathcal{P}_i, \hat{\mathcal{P}}_j). \tag{7}$$

Based on Definition 3 of imperfect representation, we rephrase the Theorem 1 to provide the following theorem:

**Theorem 2.** *Let $\hat{z}^k \in \hat{\mathcal{Z}}^k$ be the interpolated latent space between the $M_i$ and the restricted $M_j$, defined by the interpolation coefficient $\alpha$ as: $\hat{\mathcal{Z}}^k = \{\hat{z}^k = (1 - \alpha)z^i + \alpha \hat{z}^j \mid z^i \in \mathcal{Z}^i, \hat{z}^j \in \hat{\mathcal{Z}}^j\}$, where $\hat{\mathcal{P}}_k$ represents its distribution. Let the 2-Wasserstein distance between $\hat{\mathcal{P}}_k$ and $\mathcal{P}_\star$ be denoted as $W_2(\mathcal{P}_\star, \hat{\mathcal{P}}_k) = \hat{\Delta}_k$. Then the optimal $\hat{\alpha}^*$ that minimizes $\hat{\Delta}_k$ is formulated as follows: $\hat{\Delta}_k$:*

$$\hat{\alpha}^* = \frac{\Delta_i^2 - \hat{\Delta}_j^2 + \hat{\Delta}_{ij}^2}{2\hat{\Delta}_{ij}^2} \tag{8}$$

*Moreover, the resulting interpolated representation satisfies $\hat{\Delta}_k \leq \Delta_i$ and $\hat{\Delta}_k \leq \hat{\Delta}_j$.*

**Corollary 2.1.** *The optimal $\hat{\alpha}^*$ is bounded as follows:*

$$\hat{\alpha}^* = \begin{cases} \left[ \dfrac{\Delta_i}{\Delta_i + \Delta_j + \delta}, \, 1 \right] & \textit{if } \Delta_i > \hat{\Delta}_j \\[4mm] \left[ 0, \, \dfrac{\Delta_i}{\Delta_i + \Delta_j} \right] & \textit{otherwise} \end{cases} \tag{9}$$

For the proofs of Theorem 2 and Corollary 2.1, please see Appendix A.2.

**Remark 2.1.** *($\delta$ does not hinder the synergy) Although $\hat{z}^j \sim \hat{\mathcal{P}}_j$ deviates from $\mathcal{P}_j$, it still represents the latent space of the $M_j$. We can extract an imperfect feature representation from $\mathcal{P}_j$ by giving imperfect input to the modality $M_j$. This allows $\hat{z}^j$ exist in the distribution $\mathcal{P}_j$*[2]. *Consequently, $\hat{z}^j$ is closer to or part of the latent space of the $M_j$ than to that of the $M_i$ or the true latent space. Therefore, additional gap $\delta$ in Equation 6 is unlikely to significantly impact the determination of the optimal $\hat{\alpha}^*$, as $\delta$ will be generally much smaller than both $\Delta_i$ and $\Delta_j$. It stems for $\delta$ does not hinder the synergy between two modalities, i.e., the modality $M_j$ can promote the training of $M_i$, even with an imperfect representation.*

**Remark 2.2.** *($M_j$ indeed helps the training of $M_i$) In training $M_j$, we can find a superior representation $\hat{\mathcal{Z}}^k$ by utilizing imperfect feature representation from $M_j$. When revisiting the conceptual sketch in Figure 2 and the preliminary experiments on CIFAR-10 shown in the Introduction, an imperfect textual description, i.e.,* `"This is about Class #."` *works as $\hat{z}^j$ promotes the training of visual models by finding the interpolated representation.*

## 3.3 Methodology: Training Modality $M_i$ by Leveraging Modality $M_j$

Based on our theory, we thus propose a training method for one modality $M_i$ by leveraging modality $M_j$ with sampling imperfect $\hat{z}_j$ from the representation space of $M_j$. Let us consider the scenario for training modality $M_i$ by leveraging modality $M_j$, i.e., $[M_j \rightarrow M_i]$, without loss of generality.

We extend our notation by introducing an uppercase superscript to denote the data modality. Specifically, let $\mathcal{S}^i = \{(\mathbf{x}^i_m, y^i_m)\}^M_{m=1}$, representing the dataset for the $M_i$ to be learned, where $\mathbf{x}^i_m$ denotes the input data and $y^i_m$ its corresponding label, where $M$ is the number of data points sampled. Similarly, let $\mathcal{S}^j = \{\hat{\mathbf{x}}^j_m\}^M_{m=1}$ represent the sampled set of imperfect samples of the $M_j$. This yields the latent vectors $\hat{z}^j_m = g_j(\hat{\mathbf{x}}^j_m) \in \hat{\mathcal{Z}}^j$, as introduced in Definition 4, where $g_j(\cdot)$ is the pretrained model function associated with the $M_j$. $\mathcal{S}^i$ and $\mathcal{S}^j$ are sampled subsets from their respective full modality datasets, $\mathcal{D}^i$ and $\mathcal{D}^j$. In last, the primary objective is to minimize risk through Empirical Risk Minimization (ERM). We then define two associated empirical risks by the following loss functions:

$$\text{Classification Loss: } \mathcal{L}_{cls} = \mathbb{E}_{(\mathbf{x}^i_m, y^i_m) \sim \mathcal{S}^i} \left[ \mathcal{L}_{CE}(h \circ g(\mathbf{x}^i_m), y^i_m) \right]$$

$$\text{Latent Loss: } \mathcal{L}_z = \mathbb{E}_{(\mathbf{x}^i_m, y^i_m, \hat{z}^j_m) \sim \mathcal{S}^i \times \hat{\mathcal{Z}}^j} \left[ ||g(\mathbf{x}^i_m) - \hat{z}^j_m||^2 \right]$$

In this approach, the classification loss $\mathcal{L}_{cls}$ is utilized to optimize the learning of the $M_i$ while latent loss $\mathcal{L}_z$ is treated as a regularization term, enforcing alignment of features from the $M_j$.

We here propose a straightforward loss formulation, which allows to find the interpolated representations between two modalities. We linearly combine the classification and latent loss terms: $\mathcal{L}_{total} = (1 - \alpha)\mathcal{L}_{cls} + \alpha\mathcal{L}_z$, to find optimal $\hat{z}^k = g(\mathbf{x}^i)|_{\arg \min \mathcal{L}_{total}}$. A psuedocode of the learning procedures is given in Appendix B.2. Lastly, we describe how to obtain imperfect feature $\hat{z}^j$ for each modality in the following section, including vision, language, and audio.

## 4 Experimental Results

In this section, we provide an overview of the experimental results, along with detailed descriptions of the datasets, models and additional experimental settings.

### 4.1 Experimental Settings

**Datasets** For the main experiments, we test on the ImageNet-1K dataset (Krizhevsky et al., 2012) for visual tasks as the case of **[L→V]**. For further experiments in the multimodal setting, we employ the IEMOCAP (Busso et al., 2008) and AVMNIST (Liang et al., 2021; Li et al., 2023) datasets. IEMOCAP includes [A+L+Video] modalities, where we specifically focus on the [A+L] subset for our experiments. We performed experiemnts on both direction, **[L→A]** and **[A→L]**. For AVM-NIST, which contains [A+V] modalities, we followed the preprocessing steps outlined in CentralNet

---

[2]We fully describe how it can be done in the empirical testing in Section 4 and Appendix B.

Table 1: Classification results on ImageNet-1K and evaluation benchmarks (OOD and robustness)

| Model [L→V] | IN | V2 | Rend. | Sketch | A | Style. | C (↓) |
|---|---|---|---|---|---|---|---|
| ResNet-50 (reproduced) | 77.83 | 66.20 | 39.28 | 27.35 | 6.44 | 8.59 | 66.01 |
| + **BERT** (Devlin et al., 2018) | 78.41 | 67.10 | 40.38 | 28.19 | **8.47** | **9.64** | **64.96** |
| + **RoBERTa** (Liu et al., 2019) | **78.54** | **67.30** | **40.92** | **28.78** | 8.25 | 9.19 | 65.32 |
| ViT-B/32 (reproduced) | 75.04 | 62.02 | 40.31 | 27.34 | 9.23 | 16.56 | 55.45 |
| + BERT (Devlin et al., 2018) | 76.59 | 63.37 | 41.28 | 28.53 | 11.31 | 18.11 | 53.28 |
| + **RoBERTa** (Liu et al., 2019) | **76.75** | **64.00** | **41.81** | **29.50** | **11.55** | **18.75** | **52.95** |
| ViT-B/16 (reproduced) | 80.07 | 68.60 | 44.72 | 31.22 | 24.20 | 18.81 | 51.21 |
| + **BERT** (Devlin et al., 2018) | 81.62 | 70.07 | **45.72** | 33.13 | 25.12 | **20.31** | 49.27 |
| + **RoBERTa** (Liu et al., 2019) | **81.90** | **70.55** | 45.41 | **33.19** | **26.89** | 19.93 | **48.51** |

Table 2: Classification results on IEMOCAP and AVMNIST datasets on each cases of $[M_j \rightarrow M_i]$.

| Datasets | Model [L→A] | Accuracy | Model [A→L] | Accuracy |
|---|---|---|---|---|
| **IEMOCAP**[††] | Wav2Vec2[†] (Ravanelli et al., 2021) | 59.46 | BERT (Devlin et al., 2018) | 55.81 |
| | + BERT-B (Devlin et al., 2018) | 60.44 | **+ Wav2Vec2-B** (Baevski et al., 2020) | **56.49** |
| | + **BERT-L** (Devlin et al., 2018) | **61.20** | + Wav2Vec2-L (Baevski et al., 2020) | 56.05 |
| Datasets | Model [V→A] | Accuracy | Model [A→V]* | Accuracy |
| **AVMNIST** | Audio Model (Li et al., 2023) | 41.28 | Vision Model (Li et al., 2023) | 65.18 |
| | + ResNet-18 (He et al., 2016) | 42.08 | + Wav2Vec2-B (Baevski et al., 2020) | 66.37 |
| | + **ResNet-34** (He et al., 2016) | **42.44** | + **Wav2Vec2-L** (Baevski et al., 2020) | **66.69** |

[†]: SpeechBrain (Ravanelli et al., 2021) experimented with 4 out of 6 labels; we used the all labels.
[††]: Owing to transformer-type model requires numerous data, we fine-tuned the pretrained model.
*: Since the audio data in AVMNIST is based on spectrograms, we use the original raw audio data prior to its conversion into spectrogram.

(Vielzeuf et al., 2018), transforming raw audio into $112 \times 112$ spectrograms and utilizing $28 \times 28$ PCA-projected MNIST images. Similar to IEMOCAP datasets, we conducted experiments in both directions, **[V→A]** and **[A→V]**.

**Models** For the **[L→V]** case with the ImageNet-1K dataset, we employed modern architectures, including ResNet50 (He et al., 2016), ViT-B/32, and ViT-B/16 for **[V]** modality. To incorporate additional information from text prompts, we utilized pretrained BERT (Devlin et al., 2018) and RoBERTa (Liu et al., 2019), two extensively used transformer encoders for embedding the **[L]** modality. We use the "large" size versions on both encoders. For the **[L→A]** and **[A→L]** cases with IEMOCAP datasets, we employed a Wav2Vec2 (Baevski et al., 2020) model with SpeechBrain (Ravanelli et al., 2021) configurations for **[A]** modality, alongside pretrained BERT-B and BERT-L for the **[L]** modality. For the **[V→A]** case with AVMNIST experiments, we used the audio encoder from AGM (Li et al., 2023), enhanced with an additional classification layer for the **[A]** modality, and a pretrained ResNet-18 for the **[V]** modality. In addition, for the **[A→V]** case with AVMNIST, we used the original pretrained Wav2Vec2-B/L model for the **[A]** modality. Especially, in Table 2, "-B" refers to "base" size of model, while "-L" refers to "large" size configuration. For $[M_j \rightarrow M_i]$, all model parameters of $M_j$ modality are frozen.

**Additional Settings** *(how to get $\hat{z}^j$)* For the **[L→V]** case with the ImageNet-1K dataset, we obtain $\hat{z}^j$ by employing simple prompt engineering that provides imperfect and restricted information for the language modality. Specifically, prompts like `"This is about Class #."` are used, where # is a random number unrelated to the actual class label, ensuring imperfect supervision for the language modality. For the **[L→A]** case with IEMOCAP, $\hat{z}^j$ is generated using prompts like `"This is about Emotion #."`, where # is also a random number unrelated to actual labels, helping audio classification on the IEMOCAP dataset. We exclude the paired **[L]** modality data in IEMOCAP. In the **[A→L]** case with IEMOCAP, we added Gaussian noise to audio data and randomly shuffle to represent $\hat{z}^j$, thereby promoting sentiment classification in the **[L]** modality. For the **[V→A]** case with AVMNIST, we use randomly shuffled images from AVMNIST as $\hat{z}^j$ in audio classification tasks. For the **[A→V]** case in AVMNIST, $\hat{z}^j$ is generated by adding Gaussian noise to the audio data and apply shuffling, similar to the **[A→L]** case, to assist the classification task on PCA-projected MNIST images. Detail description of implementing $\hat{z}^j$ is provided on Table 5.

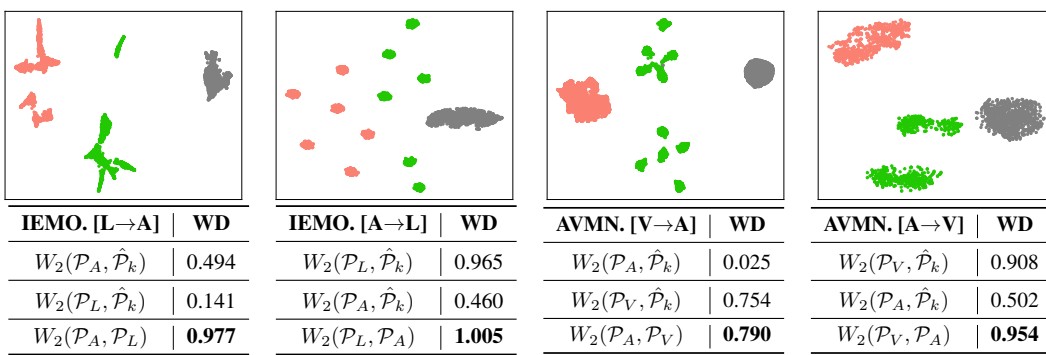

| IEMO. [L→A] | WD | IEMO. [A→L] | WD | AVMN. [V→A] | WD | AVMN. [A→V] | WD |
|---|---|---|---|---|---|---|---|
| $W_2(\mathcal{P}_A, \hat{\mathcal{P}}_k)$ | 0.494 | $W_2(\mathcal{P}_L, \hat{\mathcal{P}}_k)$ | 0.965 | $W_2(\mathcal{P}_A, \hat{\mathcal{P}}_k)$ | 0.025 | $W_2(\mathcal{P}_V, \hat{\mathcal{P}}_k)$ | 0.908 |
| $W_2(\mathcal{P}_L, \hat{\mathcal{P}}_k)$ | 0.141 | $W_2(\mathcal{P}_A, \hat{\mathcal{P}}_k)$ | 0.460 | $W_2(\mathcal{P}_V, \hat{\mathcal{P}}_k)$ | 0.754 | $W_2(\mathcal{P}_A, \hat{\mathcal{P}}_k)$ | 0.502 |
| $W_2(\mathcal{P}_A, \mathcal{P}_L)$ | **0.977** | $W_2(\mathcal{P}_L, \mathcal{P}_A)$ | **1.005** | $W_2(\mathcal{P}_A, \mathcal{P}_V)$ | **0.790** | $W_2(\mathcal{P}_V, \mathcal{P}_A)$ | **0.954** |

Figure 3: t-SNE visualizations and Wasserstein Distances (WD) across multimodal datasets

**Implementation** For the **[L→V]** case in ImageNet-1K classification task, we followed hyperparameter settings from the AugReg-ViT (Steiner et al., 2022) in training ResNet50, ViT-B/32, and ViT-B/16. For multimodal datasets, i.e., IEMOCAP and AVMNIST, particularly in the cases of **[L→A]**, **[A→L]**, **[V→A]** and **[A→V]**, we employed customized hyperparameter settings for each case. We consistently used 30 training epochs with Adam optimizer (Kingma & Ba, 2015). We omitted any data augmentations. Lastly, we applied $\alpha = 0.5$ for the **[L→V]** case, and $\alpha = 0.3$ for the other cases. Additional details are provided in Appendix B.

## 4.2 RESULTS

**Main Results** Our main results are two parts: Table 1 with ImageNet-1K and Table 2 with multimodal datasets, i.e., IEMOCAP and AVMNIST.

According to Table 1, the results for the multimodal learning case, especially **[L→V]** case, highlights the outstanding performance of our approach on ImageNet-1K (IN). where it achieved improvements between approximately $+1.5\%$ and $+2.0\%$. Notably, our approach also demonstrates significant improvements across additional evaluation datasets, including ImageNet-V2, out-of-distribution (OOD) datasets such as ImageNet-Rend. (Hendrycks et al., 2020), ImageNet-Sketch (Wang et al., 2019), and ImageNet-Style. (Geirhos et al., 2019), achieving performance gains between $+1.0\%$ and $+2.0\%$. Furthermore, our approach also excels in robustness benchmarks, achieving improvements up to $+2.6\%$ on adversarial examples from ImageNet-A (Hendrycks et al., 2020) and 2.5↓ on corrupted images from ImageNet-C (Hendrycks & Dietterich, 2019). Surprisingly, language models clearly promote the training of visual models, i.e., writers indeed help painters. Notably, the synergy shows consistent gains even in OOD, adversarial, and corrupted samples. It emphasizes that language models also facilitate visual models to acquire representations, which are well-generalized on a wide range of visual data distributions.

For more cases with IEMOCAP and AVMNIST in Table 2, our approach also leads to performance gains across various tests. In the IEMOCAP experiments, it shows a considerable gain of approximately $+1.6\%$ in the **[L→A]** case and $+0.6\%$ gain in the **[A→L]** case. For the AVMNIST testing, the performance improvement is around $+1.2\%$ in the **[A→L]** case and $+1.4\%$ in the **[A→L]** case.

Consequently, we confirm that the three modalities—**V**ision, **L**anguage, and **A**udio—are shown to mutually enhance each other's training in ways that are not easily anticipated. This underscores the potential for significant performance improvements when multiple modalities are effectively integrated and demonstrates the broad applicability of our claims across different types of modalities. Moreover, we emphasize that our claims have been validated across numerous existing deep model architectures, where this validation highlights the model-agnostic nature of our approach, making it applicable to a broad spectrum of scenarios.

## 5 ANALYSIS

### 5.1 ANALYSIS AND REPRESENTATION VISUALIZATIONS

**Wasserstein Distance Between Modalities** Our hypothesis suggests that the Wasserstein distance between the latent feature distributions of individual modalities should exceed that of an adaptively

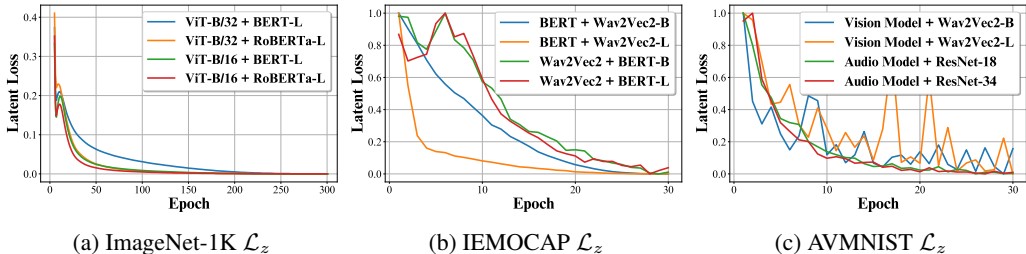

Figure 4: Validation Latent loss $\mathcal{L}_z$ across the entire datasets

Table 3: Evaluation across the different value of $\alpha$

| Datasets | Accuracy | | | | |
|---|---|---|---|---|---|
| | $\alpha = 0$ | $\alpha = 0.1$ | $\boldsymbol{\alpha = 0.3}$ | $\alpha = 0.5$ | $\alpha = 0.7$ |
| IEMOCAP [L→A] | 59.46 | 60.34 | **61.20** | 59.87 | 59.93 |
| IEMOCAP [A→L] | 55.81 | 55.51 | **56.49** | 54.19 | 55.90 |
| AVMNIST [V→A] | 41.28 | 42.03 | **42.44** | 41.77 | 41.79 |
| AVMNIST [A→V] | 65.18 | 65.03 | **66.69** | 64.77 | 64.76 |

Table 4: Comparison: $\hat{z}^j$ vs. $z^j$

| Model [L→V] | $\hat{z}^j$ | $z^j$ |
|---|---|---|
| ResNet-50 + RoBERTa | 78.54 | 78.61 (+0.07) |
| ViT-B/32 + RoBERTa | 76.75 | 76.99 (+0.24) |
| ViT-B/16 + RoBERTa | 81.90 | 82.54 (+0.64) |

trained distribution, $\hat{\mathcal{P}}_k$, which ideally lies between the latent spaces of the given modalities. Figure 3 illustrates this concept. In the test of $[M_j \rightarrow M_i]$ scenario, the ▬ denote latent vectors $z^i$ from the $M_i$ modality, ▬ represent $\{\hat{z}_m^k\}_{m=1}^M$, and ▬ represent $M_j$ modality latent vectors $z^j$. As expected, t-SNE visualizations largely represent that $\hat{z}_m^k$ resides between the latent spaces of $M_i$ and $M_j$, with asmaller Wasserstein distance to each modality (see $W_2(\cdot, \hat{\mathcal{P}}_k)$ terms), while individual modalities are shown to be distant to each other. These findings visually illustrate our hypothesis, coinciding with the claim of 'interpolated' representations. It further confirms that our $\mathcal{L}_{total}$ empirically serves as an adequate objective function to acquire the superior interpolate representation in between individual modalities.

**The Convergence of Latent Loss** We here show how the latent loss, i.e., $\mathcal{L}_z$, which reflects how much the interpolated representation moves close to the modality $M_j$, behaves during the training. As shown in Figure 4, the loss consistently decreases throughout all cases. This result highlights the narrowing gap between the latent vectors from modality $M_j$ and the representation vectors derived from the input data. By minimizing this gap, models successfully exploits simplified yet informative latent features $\hat{z}^j$, thereby improving overall performance. As Mean Squared Error (MSE) is employed, the magnitude of the gap may vary across experiments due to dimensions of representation vectors. To account for this, we applied normalized latent loss to observe a clear convergence for all cases. Despite some fluctuations in certain cases, the loss consistently decreases and converges to a value close to, but not exactly, zero due to interpolation.

## 5.2 ABLATION STUDIES

**The Effect of $\alpha$** Due to the fact that $\alpha$ plays a crucial role in finding optimal latent space, we evaluated how variations in $\alpha$ influence performance on multimodal datasets. We tested various values of $\alpha = \{0.0, 0.1, 0.3, 0.5, 0.7\}$, and the corresponding results are shown in Table 3. We observe clear performance degradation when a biased $\alpha$ with too small or large values is used. It demonstrates that the biased cases, which tend to strongly rely on one single modality, do not show a synergy. Among candidates, $\alpha = 0.3$ consistently shows the best accuracies, highlighting the importance of balancing the contribution of each modality. When reminding t-SNE visualizations in Figure 3, $\alpha$ with a moderate value coincides with the spatial position of the representation at the middle of two individual representations.

**The Usage of Paired Supervision $\mathcal{Z}^j$ vs. $\hat{\mathcal{Z}}^j$:** Rather than using imperfect $\hat{z}^j$, we investigate how much gains would be further achieved when using the perfectly matched $z^j$. In Table 4, we tested the ImageNet-1K (IN) cases, where the column with '$\hat{z}^j$' refers to the numbers in Table 1 of the main experiments, and the column with '$z^j$' is done with perfectly matched supervision in the **[L]**

modality.[3] As shown in Table 4, although the performance slightly improves, the gains are minimal. This result aligns with our hypothesis, indicating that $\delta$ from Theorem 2 has a limited effect on performance, supporting our hypothesis. Also, the result emphasizes that our approach significantly relieves the cost of pairing perfectly matched supervision across modalities while achieving a comparative performance with the ideal case with paired supervision.

## 6  FURTHER DISCUSSIONS AND LIMITATIONS

**Innovating Multimodal Learning:** In prior works, many multimodal learning methods have relied on paired-datasets for training. This contrasts with human learning, which often occurs without the need for precisely paired object from different modality. Similar to human's perspective, our work seek to overcome this limitation, demonstrating that minimal or even imperfect supervision from different modalities can still enhance learning in the primary modality. Consequently, our approach suggest further advancements in technical perspective, enabling the effective utilization of multimodal settings even when only a single modality is available.

**Discovery of Unexpected Synergy Between Modalities:** It is commonly assumed that not every modality can effectively assist another. The prevailing notion is that unrelated modalities may not provide meaningful assistance to one another. However, our experiments on **[V→A]** and **[A→V]** reveal that cross-modality interactions, even between seemingly unrelated domains, can lead to significant performance improvements. These results demonstrate that integrating seemingly unrelated modalities can still yield benefits, uncovering hidden correlations and unexpected synergies between them. This approach opens new possibilities for exploring multimodal combinations that were previously considered non-beneficial, such as integrating language with signal-based sensory data. Expanding the scope of multimodal learning may reveal hidden synergies between multiple modalities, leading to improved overall performance.

**Broad Impacts on Wide Range of Tasks:** Recent works has focused on addressing the generalization problem, such as out-of-distribution (OOD), adverserial, and robustness. We validated that our method not only achieves strong performance on the ImageNet-1K but also shows significant improvements in OOD, adversarial, and robustness scenarios, as shown in Table 1. These findings suggest that our approach could be extended to further tackle the generalization problem by incorporating additional guidance from imperfect data across other modalities.

**Limitations:** While our experiments demonstrate the effectiveness of the proposed approach, certain limitations remain. One key challenge is scalability, both in terms of computational feasibility, and the other is theoretical extensions to multiple modalities.

Primarily, due to resource constraints, it is limited to conduct large-scale evaluations on larger architecture size, leaving empirical model scalability of our method an open challenge for future research. However, it is important to emphasize that our theoretical framework is not inherently restricted by model size. We expect the core principles of our approach to generalize effectively to larger models.

The other limitation, as mentioned on the previous discussion, is that our method has been evaluated only on the paired modalities. Extending it to incorporate more than two modalities could unlock further performance gains for individual modalities. Nonetheless, aligning latent vector dimensions across multiple modalities introduces additional complexity, which may impact performance and requires further investigation.

## 7  CONCLUSIONS

We presented both theoretical and empirical frameworks demonstrating that one modality with imperfect representation can still enhance learning in the other modality. Our results and extensive analyses support proposed hypotheses and reinforce theoretical foundations of this approach. Notably, we showed that paired supervision between datasets is unnecessary, as weakly related supervision or even mismatched setting across modalities can still lead to improvements. In last, we then propose future research to explore more complex multimodal settings, such as leveraging more than two modalities or scaling to larger models, which may drive significant advancements in this field.

---

[3]Additional details providing more precise information are included in the Appendix. B.

## ACKNOWLEDGMENTS

This work was supported by the Institute of Information & communications Technology Planning & Evaluation (IITP) grant funded by the Korea government (MSIT) (No. RS-2020-II201336, Artificial Intelligence Graduate School Program (UNIST)), (No. IITP-2025-RS-2022-00156361, Innovative Human Resource Development for Local Intellectualization program), and the National Research Foundation of Korea (NRF) grant funded by the Korea government (MSIT) (No. RS-2024-00459023).

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

## A  MATHEMATICAL DETAILS FOR THEORETICAL FRAMEWORKS

### A.1  PROOF OF THEOREM 1 AND COROLLARY 1.1

**Theorem 1.** *The optimal $\alpha^*$ that minimizes $\Delta_k$ is formulated as follows:*

$$\alpha^* = \frac{\Delta_i^2 - \Delta_j^2 + \Delta_{ij}^2}{2\Delta_{ij}^2}$$

*Moreover, the resulting interpolated representation satisfies $\Delta_k \leq \Delta_i$ and $\Delta_k \leq \Delta_j$.*

*Proof.* Given that Wasserstein distance is symmetric and satisfies triangle inequality (Villani et al., 2009; Peyré & Cuturi, 2019), we can define the relationship each distribution of latent spaces:

$$W_2(\mathcal{P}_i, \mathcal{P}_j) \leq W_2(\mathcal{P}_\star, \mathcal{P}_i) + W_2(\mathcal{P}_\star, \mathcal{P}_j) \tag{10}$$

and this could be simplified via notations provided on Definitions at Section 3.2:

$$\Delta_{ij} \leq \Delta_i + \Delta_j \tag{11}$$

According to prior works (Villani et al., 2009; Peyré & Cuturi, 2019; Mahey et al., 2024), we can define a new equation as follows about the relation between wasserstein distances:

**Definition 5** (**Generalized Geodesics** (Mahey et al., 2024)). *According to the convexity property of 2-Wasserstein distance, squared 2-Wasserstein distance between $\mathcal{P}_k$ and $\mathcal{P}_\star$ is bounded by:*

$$W_2^2(\mathcal{P}_k, \mathcal{P}_\star) \leq (1-\alpha)W_2^2(\mathcal{P}_\star, \mathcal{P}_i) + \alpha W_2^2(\mathcal{P}_\star, \mathcal{P}_j) - \alpha(1-\alpha)W_2^2(\mathcal{P}_i, \mathcal{P}_j) \tag{12}$$

$$= (1-\alpha)\Delta_i^2 + \alpha\Delta_j^2 - \alpha(1-\alpha)\Delta_{ij}^2 \tag{13}$$

Considering that our primary goal is to determine the optimal $\alpha$, we can express Equation 13 as the quadratic function of $\alpha$, i.e., $f(\alpha) = (1-\alpha)\Delta_i^2 + \alpha\Delta_j^2 - \alpha(1-\alpha)\Delta_{ij}^2$. The optimal value of $\alpha$ can then be determined by taking the derivative of the function $f(\alpha)$:

$$\frac{\partial f(\alpha)}{\partial \alpha} = -\Delta_i^2 + \Delta_j^2 - (1-2\alpha)\Delta_{ij}^2 = 0 \tag{14}$$

Changing into the term of $\alpha$ is then :

$$\alpha^* = \frac{1}{2}\left(1 + \frac{\Delta_i^2 - \Delta_j^2}{\Delta_{ij}^2}\right) = \frac{1+C}{2} \quad \text{where} \quad C = \frac{(\Delta_i + \Delta_j)(\Delta_i - \Delta_j)}{\Delta_{ij}^2} \tag{15}$$

where $\alpha^* = \arg\min_\alpha f(\alpha)$. This optimal $\alpha^*$ remains valid under the Assumption 1 that $\Delta_{ij}$ is bigger than 0, ensuring that $\alpha^*$ does not diverge.

It still remains to be proven that the optimal value of $\alpha^*$ indeed minimizes $\Delta_k$ and satisfies the condition that $\Delta_k$ is smaller than both $\Delta_i$ and $\Delta_j$. To establish this, we begin by verifying if $\Delta_k$ is truly minimized. Let the minimized value of $\Delta_k$ be denoted as $\Delta_k^*$. Assume that $z^k \sim \mathcal{P}_k$ lies on the geodesic between $z^i \sim \mathcal{P}_i$ and $z^j \sim \mathcal{P}_j$, representing a projection onto the true latent space.[4]

To demonstrate this, we can apply the Pythagorean theorem. Under the above assumption, $\Delta_k^*$ can be expressed as:

$$\Delta_k^{*2} = \Delta_i^2 - (\alpha\Delta_{ij})^2 \tag{16}$$

$$= \Delta_j^2 - ((1-\alpha)\Delta_{ij})^2 \tag{17}$$

We can further rearrange the above equations as:

$$\Delta_i^2 - \Delta_j^2 = (2\alpha - 1)\Delta_{ij}^2 \tag{18}$$

---

[4]This concept is detailed in the min-SWDD paper (Mahey et al., 2024).

From this, we solve for optimal $\alpha$, denoted as $\alpha^*$:

$$\alpha^* = \frac{1}{2}\left(1 + \frac{\Delta_i^2 - \Delta_j^2}{\Delta_{ij}^2}\right) \tag{19}$$

This expression corresponds to the optimal value $\alpha^*$ from Equation 4 and satisfies the Pythagorean theorem, which we assumed to hold when minimizing $\Delta_k$. This minimized value of $\Delta_k$ represents the distance between $\mathcal{P}_\star$ and the distribution $\mathcal{P}_k$, projected directly onto the geodesic connecting $\mathcal{P}_i$ and $\mathcal{P}_j$. Moreover, we conclude that the minimized $\Delta_k^*$ is indeed smaller than both $\Delta_i$ and $\Delta_j$, where $\Delta_k^{*2} \leq \Delta_i^2$ as related to Equation 16 and $\Delta_k^{*2} \leq \Delta_j^2$ as related to Equation 17. Therefore, result is formally expressed as:

$$\Delta_k^* \leq \min(\Delta_i, \Delta_j) \tag{20}$$

where it satisfies the Theorem 1. □

**Corollary 1.1** *The optimal $\alpha^*$ is bounded as follows:*

$$\alpha^* = \begin{cases} \left[\dfrac{\Delta_i}{\Delta_i + \Delta_j},\ 1\right] & \text{if } \Delta_i > \Delta_j \\[3mm] \left[0,\ \dfrac{\Delta_i}{\Delta_i + \Delta_j}\right] & \text{otherwise} \end{cases}$$

*Proof.* To establish the validity of the optimal $\alpha^*$ under different conditions for $\Delta_i$ and $\Delta_j$, we begin by examining the relationship between these two quantities.

- **Condition 1.1: $(\Delta_i > \Delta_j)$** : This scenario implies that the difference between $\Delta_i$ and $\Delta_j$ is positive, meaning $C = \frac{\Delta_i - \Delta_j}{\Delta_i + \Delta_j}$ exhibits a positive value. In this case, the constant $C$ is bounded by the inequality:

$$C \geq \frac{\Delta_i - \Delta_j}{\Delta_i + \Delta_j} \tag{21}$$

Substituting this bound into Equation 15, we have the following inequality for the optimal value of $\alpha^*$:

$$\alpha^* \geq \frac{1}{2}\left(1 + \frac{\Delta_i - \Delta_j}{\Delta_i + \Delta_j}\right) \tag{22}$$

$$= \frac{\Delta_i}{\Delta_i + \Delta_j} \tag{23}$$

As we know that $\alpha^*$ is constrained by $0 \leq \alpha^* \leq 1$, this inequality confirms that $\alpha^* = \frac{\Delta_i}{\Delta_i + \Delta_j}$ satisfies the first condition of optimality, as given by Equation 5. This ensures that when $\Delta_i > \Delta_j$, the chosen value of $\alpha^*$ lies within the allowable range and maintains the necessary relationship between the distances.

- **Condition 1.2: $(\Delta_i \leq \Delta_j)$** : In this case, the difference between $\Delta_i$ and $\Delta_j$ is non-positive, and consequently, the value of $C$ becomes negative. Thus, the constant $C$ is bounded by the inequality:

$$C \leq \frac{\Delta_i - \Delta_j}{\Delta_i + \Delta_j} \tag{24}$$

By applying the same process as in Condition 1, we substitute this bound into the formula for $\alpha^*$, yielding:

$$\alpha^* \leq \frac{\Delta_i}{\Delta_i + \Delta_j} \tag{25}$$

Given that $\alpha^*$ must satisfy $0 \leq \alpha^* \leq 1$, this condition is similarly met. Moreover, as $\alpha^* \geq 0$, this value conforms to the second condition of optimality in Equation 5. Hence, when $\Delta_i \leq \Delta_j$, the optimal value of $\alpha^*$ continues to fulfill the constraints of the inequality, ensuring that the geometric relationship between the distances is preserved.

Thus, the optimal value $\alpha^*$ satisfies both conditions for the two possible relationships between $\Delta_i$ and $\Delta_j$, concluding the proof. □

### A.2 Proof of Theorem 2 and Corollary 2.1

**Theorem 2.** *Let $\hat{z}^k \in \hat{\mathcal{Z}}^k$ be the interpolated latent space between the $M_i$ and the restricted $M_j$, defined by the interpolation coefficient $\alpha$ as: $\hat{\mathcal{Z}}^k = \{\hat{z}^k = (1-\alpha)z^i + \alpha\hat{z}^j \mid z^i \in \mathcal{Z}^i, \hat{z}^j \in \hat{\mathcal{Z}}^j\}$, where $\hat{\mathcal{P}}_k$ represents its distribution. Let the 2-Wasserstein distance between $\hat{\mathcal{P}}_k$ and $\mathcal{P}_\star$ be denoted as $W_2(\mathcal{P}_\star, \hat{\mathcal{P}}_k) = \hat{\Delta}_k$. Then the optimal $\hat{\alpha}^*$ that minimizes $\hat{\Delta}_k$ is formulated as follows: $\hat{\Delta}_k$:*

$$\hat{\alpha}^* = \frac{\Delta_i^2 - \hat{\Delta}_j^2 + \hat{\Delta}_{ij}^2}{2\hat{\Delta}_{ij}^2}$$

*Moreover, the resulting interpolated representation satisfies $\hat{\Delta}_k \leq \Delta_i$ and $\hat{\Delta}_k \leq \hat{\Delta}_j$.*

*Proof.* Similar to the approach in Appendix A.1, $\hat{\Delta}_k$ can be defined with following Definition 5:

$$W_2^2(\mathcal{P}_\star, \hat{\mathcal{P}}_k) \leq (1-\alpha)W_2^2(\mathcal{P}_\star, \mathcal{P}_i) + \alpha W_2^2(\mathcal{P}_\star, \hat{\mathcal{P}}_j) - \alpha(1-\alpha)W_2^2(\mathcal{P}_i, \hat{\mathcal{P}}_j) \qquad (26)$$

$$= (1-\alpha)\Delta_i^2 + \alpha\hat{\Delta}_j^2 - \alpha(1-\alpha)\hat{\Delta}_{ij}^2 \qquad (27)$$

To determine the optimal $\alpha$ that minimizes the left-hand side of Equation 27, we can utilize the derivative of the function $f(\alpha) = (1-\alpha)\Delta_i^2 + \alpha\hat{\Delta}_j^2 - \alpha(1-\alpha)\hat{\Delta}_{ij}^2$ and set it equal to zero, as demonstrated in Equation 14:

$$\frac{\partial f(\alpha)}{\partial \alpha} = -\Delta_i^2 + \hat{\Delta}_j^2 - (1-2\alpha)\hat{\Delta}_{ij}^2 = 0. \qquad (28)$$

This can be rephrased in terms of $\alpha$, analogous to Equation 15:

$$\hat{\alpha}^* = \frac{1}{2}\left(1 + \frac{\Delta_i^2 - \hat{\Delta}_j^2}{\hat{\Delta}_{ij}^2}\right) = \frac{1+C'}{2}, \quad \text{where } C' = \frac{(\Delta_i + \hat{\Delta}_j)(\Delta_i - \hat{\Delta}_j)}{\hat{\Delta}_{ij}^2} \qquad (29)$$

Additionally, the optimal $\hat{\alpha}^*$ is valid, similar to the validity of $\alpha^*$, under Assumption 1 and the property of $\hat{\Delta}_{ij}$, where $\Delta_{ij} \leq \hat{\Delta}_{ij}$.

To achieve the goal of minimizing $\hat{\Delta}_k$, we need to demonstrate that applying the optimal $\hat{\alpha}^*$ results in a value of $\hat{\Delta}_k$ smaller than both $\Delta_i$ and $\hat{\Delta}_j$. Following the previous steps outlined in the proof of Theorem 1, we can proceed as follows.

Let the minimized distance be denoted as $\hat{\Delta}_k^*$, with the optimal coefficient being $\hat{\alpha}^*$. Using the Pythagorean Theorem, we relate the distances $\Delta_i$, $\hat{\Delta}_j$, and $\hat{\Delta}_{ij}$ through the following expressions:

$$\hat{\Delta}_k^{*2} = \Delta_i^2 - (\hat{\alpha}^*\hat{\Delta}_{ij})^2 \qquad (30)$$

$$= \hat{\Delta}_j^2 - ((1-\hat{\alpha}^*)\hat{\Delta}_{ij})^2 \qquad (31)$$

These two equations can be rearranged in terms of $\hat{\alpha}^*$, leading to the expression:

$$\Delta_i^2 - \hat{\Delta}_j^2 = (2\hat{\alpha}^* - 1)\hat{\Delta}_{ij}^2 \qquad (32)$$

From this, solving for $\hat{\alpha}^*$ yields:

$$\hat{\alpha}^* = \frac{1}{2}\left(1 + \frac{\Delta_i^2 - \hat{\Delta}_j^2}{\hat{\Delta}_{ij}^2}\right) \qquad (33)$$

This expression is equivalent to Equation 8 in Theorem 2. The assumption that the optimal $\hat{\alpha}^*$ minimizes $\hat{\Delta}_k^*$, which denotes the geodesic projection of the distribution $\hat{\mathcal{P}}_k$ between $\mathcal{P}_i$ and $\hat{\mathcal{P}}_j$, aligns with the closed-form solution for $\hat{\alpha}^*$ given in Definition 5 and Equation 29.

Furthermore, by substituting into Equations 30 and 31, the minimization of $\hat{\Delta}_k^2$ implies:

$$\hat{\Delta}_k^2 \leq \min(\Delta_i^2, \hat{\Delta}_j^2) \tag{34}$$

$$\Rightarrow \hat{\Delta}_k \leq \min(\Delta_i, \hat{\Delta}_j) \tag{35}$$

Based on Definition 4 and the triangle inequality, $\hat{\Delta}_j$ can be further bounded as $\hat{\Delta}_j \leq \Delta_j + \delta$. However, as discussed in Remark 2.1, the term $\delta$ is expected to have a negligible effect on the Wasserstein distance between the distributions, resulting in only a minor constant offset. Therefore, this does not significantly impact the Wasserstein distance between the distributions, which Equation 35 will approximately similar to Equation 20. □

**Corollary 2.1** *The optimal $\hat{\alpha}^*$ is bounded as follows:*

$$\hat{\alpha}^* = \begin{cases} \left[ \dfrac{\Delta_i}{\Delta_i + \Delta_j + \delta}, \, 1 \right] & \textit{if } \Delta_i > \hat{\Delta}_j \\[4mm] \left[ 0, \, \dfrac{\Delta_i}{\Delta_i + \Delta_j} \right] & \textit{otherwise} \end{cases}$$

*Proof.* We analyze the bounds on $\hat{\alpha}^*$ by considering two conditions based on $\Delta_i$ and $\hat{\Delta}_j$:

- **Condition 2.1 ($\boldsymbol{\Delta_i > \hat{\Delta}_j}$):** In this case, $C'$ is positive. From the triangle inequality for the 2-Wasserstein distance, we obtain the following lower bound:

$$C' \geq \frac{\Delta_i - \hat{\Delta}_j}{\Delta_i + \hat{\Delta}_j} \tag{36}$$

Substituting this bound into Equation 29, we derive:

$$\hat{\alpha}^* \geq \frac{\Delta_i}{\Delta_i + \hat{\Delta}_j} \tag{37}$$

$$\geq \frac{\Delta_i}{\Delta_i + \Delta_j + \delta} \tag{38}$$

where $\delta$ is a small constant, accounting for minor deviations between $\hat{\Delta}_j$ and $\Delta_j$, as discussed in Remark 2.1. Since $\hat{\alpha}^* \leq 1$, this satisfies the first condition in Equation 9.

- **Condition 2.2 ($\boldsymbol{\Delta_i \leq \hat{\Delta}_j}$):** In this scenario, $C'$ becomes negative. Using a similar approach as in Case 1, the upper bound for $C'$ is:

$$C' \leq \frac{\Delta_i - \hat{\Delta}_j}{\Delta_i + \hat{\Delta}_j} \tag{39}$$

Substituting this into Equation 29, we get:

$$\hat{\alpha}^* \leq \frac{\Delta_i}{\Delta_i + \hat{\Delta}_j} \tag{40}$$

Since $\hat{\Delta}_j \geq \Delta_j$, we can substitute $\hat{\Delta}_j$ with $\Delta_j$, resulting in a looser bound:

$$\hat{\alpha}^* \leq \frac{\Delta_i}{\Delta_i + \Delta_j} \tag{41}$$

Thus, this bound satisfies the second condition of Equation 9.

Therefore, we find that the derived bounds on $\hat{\alpha}^*$ fulfill the both conditions stated in Equation 9. □

## B   IMPLEMENTATION DETAILS

### B.1   DATASETS

**ImageNet-1K and Evaluation Benchmarks** ImageNet-1K (Krizhevsky et al., 2012) is the image datasets that contains 1000 classes with 1,281,167 training images and 50,000 validation images. ImageNet has been widely used in image classification benchmarks in various methods, especially in computer vision task (Dosovitskiy et al., 2021; Steiner et al., 2022; Zhou et al., 2023; Pang et al., 2024). In our evaluation, we also assessed ImageNet-related validation benchmarks focusing on out-of-distribution (OOD) and robustness scenarios. These benchmarks can be categorized into three types: in-domain, OOD, and robustness.

For the in-domain category, we utilized ImageNet-V2 (Recht et al., 2019), which consists of 10 images per class from the original ImageNet-1K, with total 10,000 images. The OOD benchmarks contains ImageNet-Rendition (Hendrycks et al., 2020), ImageNet-Sketch (Wang et al., 2019), and ImageNet-Stylized (Geirhos et al., 2019). ImageNet-Rendition features 200 classes with a total of 30,000 images, while ImageNet-Sketch contains approximately 50 images per class, totaling 50,889 images sourced from Google image queries labeled as `"sketch of {class name}."` Lastly, the robustness scenarios encompass ImageNet-A (Hendrycks et al., 2020) and ImageNet-C (Hendrycks & Dietterich, 2019). ImageNet-A consists of images misclassified by the ResNet-50 model. ImageNet-C features a variety of generated corruptions, such as Gaussian noise and blurring, and is commonly used in adversarial training approaches.

**IEMOCAP** The IEMOCAP dataset (Busso et al., 2008) contains video, language and audio modalities. It consists of 151 recorded dialogue videos featuring two speakers per session, resulting in a total of 302 videos. Each segment is annotated for nine emotions: angry, excited, fear, sad, surprised, frustrated, happy, disappointed, and neutral. Recorded across five sessions with five pairs of speakers. It also contains the audio and script for each video data. IEMOCAP dataset is a valuable resource for research in multimodal emotion recognition and has been widely employed in various multimodal methods (Zhang et al., 2020; Li et al., 2018) for the emotion sentiment classification.

As mentioned in Section 4.1, our implementation focused solely on the script and audio components, emphasizing the language and audio modalities. For convenience, we narrowed our analysis to six specific emotions: neutral, happy, angry, sad, excited, and frustrated, employing the official PyTorch framework (Paszke et al., 2019) for dataloader.

**AVMNIST** The AVMNIST dataset (Vielzeuf et al., 2018) contains digit images (0 to 9) from the MNIST dataset (Lecun et al., 1998), where each image has dimensions of $28 \times 28 \times 1$. These images have been subjected to PCA projection, resulting in a reduced information representation compared to the original MNIST dataset. In addition to the visual modality, the dataset includes an audio modality from the Free Spoken Digits Dataset (FSDD) (Jackson et al., 2018). The audio data has been preprocessed into mel-frequency spectrograms, sampled at a resolution of $112 \times 112 \times 1$.

In the context of the audio modality [A], we approach the two cases, **[V→A]** and **[A→V]**, using distinct strategies. For the **[V→A]** case, the audio modality is processed using convolutional neural networks (CNNs), outlined in AGM (Li et al., 2023). In contrast, for the **[A→V]** case, we utilize raw audio data from the FSDD dataset rather than mel-spectrogram images, as pretrained models specifically trained on spectrogram data are not readily available for this task.

**Representation of $z^j$** In Section 5.2, we discussed the representation of $z^j$. As ImageNet-1K does not contains any supervision related to the **[L]** modality, except class label, we employed a new description for each image generated by LLaVA (Liu et al., 2024).

Specifically, we input images from ImageNet along with the following prompt: `"USER: <image>\nWhat does this image represent? Explain in a sentence.\nASSISTANT:".` LLaVA generates a descriptive prompt for each image, which provides a description that is closer to the true supervision of the language modality, $z^j$, compared to our original prompt, `"This is a class about #.",` $\hat{z}^j$. As demonstrated in Table 4, this change does not result in significant performance differences, where it aligns to our theoretical perspective in Theorem 2 and Remark 2.1 that $\delta$ does not substantially affect the distance between the distributions of each modality.

## B.2 EXPERIMENTAL DETAILS

**Pseudo Code and Detail Implementation of $\hat{z}^j$** We provide the pseudo-code on Algorithm 1 and implementation details for $\hat{z}_m^j$ in all cases in Table 5 as follows:

Table 5: Implementation details of $\hat{z}_m^j$ across different datasets and cases

| Datasets & Cases | Implementation of $\hat{z}_m^j$ |
|---|---|
| ImageNet-1k [**L**→**V**] | [**L**] $\Rightarrow$ `This is about Class #.`[†] |
| IEMOCAP [**L**→**A**] | [**L**] $\Rightarrow$ `This is about Emotion #.`[†] |
| IEMOCAP [**A**→**L**] | [**A**] $\Rightarrow$ Add Gaussian Noise: $\xi \sim \mathcal{N}(0, \lambda I)$[††] & Random Shuffling |
| AVMNIST [**V**→**A**] | [**V**] $\Rightarrow$ Random Shuffled Image (mismatch paired sets) |
| AVMNIST [**A**→**V**] | [**A**] $\Rightarrow$ Add Gaussian Noise: $\xi \sim \mathcal{N}(0, \lambda I)$[††] & Random Shuffling |

[†]: # is a random number that does not directly correspond to the actual label.
[††]: $\lambda$ is a parameter that controls the variance of the Gaussian noise. We applied $\lambda = 10^{-3}$

---

**Algorithm 1** Traininig Procedures for $[M_j \rightarrow M_i]$

---

**Hyperparameter:** $\alpha$: interpolate coefficients, $\mathcal{B}$: batch size

**Input:** $\{(x_m^i, y_m^i)\}_{m=1}^M \sim \mathcal{S}^i$: input data from $M_i$ modality sampled in batch size $\mathcal{B}$ for each, $\{\hat{x}_m^j\}_{m=1}^M \sim \mathcal{S}^j$: imperfect input data from $M_j$ modality sampled in batch size $\mathcal{B}$ for each

**Required:** $\mathcal{L}_{CE}$: Cross-Entropy Loss, $\theta_i$: $M_i$ modality model parameters

**Function:** $g_i(\cdot; \theta_i)$: latent feature mapping function of $M_i$ modality, $g_j(\cdot; \theta_j)$: latent feature mapping function of $M_j$ modality, $h_i(\cdot; \theta_i)$: hypothesis function of $M_i$ modality

1: **while** not done **do**
2:     **for** $m = 1, \ldots, M$ **do**
3:         $\hat{z}_m^j = g_j(\hat{x}_m^j; \theta_j)$
4:         $\mathcal{L}_{cls} = \mathcal{L}_{CE}(h_i(g_i(\mathbf{x}_m^i; \theta_i); \theta_i), y_m^i)$
5:         $\mathcal{L}_z = \mathbb{E}\left[||g_i(\mathbf{x}_m^i; \theta_i) - \hat{z}_m^j||_2^2\right]$
6:         $\mathcal{L}_{total} = (1 - \alpha)\mathcal{L}_{cls} + \alpha\mathcal{L}_z$
7:         $\theta_i \leftarrow \theta_i - \nabla_{\theta_i}\mathcal{L}_{total}$
8:     **end for**
9: **end while**

---

**Hyperparameters Settings** In the [**L**→**V**] case for the ImageNet-1K classification task, we adhered to the hyperparameter settings established by AugReg-ViT (Steiner et al., 2022) for all training models, specifically ResNet-50, ViT-B/32, and ViT-B/16. For the baseline model, we trained for 300 epochs with a batch size of 1024, utilizing a learning rate of $1 \times 10^{-3}$ and a weight decay of $5 \times 10^{-2}$. We employed the AdamW optimizer (Loshchilov & Hutter, 2019) with cosine learning rate scheduling (Loshchilov & Hutter, 2017) and implemented a linear warmup for 20 epochs. Data augmentations, including MixUp (Zhang et al., 2018), RandAugment (Cubuk et al., 2020), and AugReg (Steiner et al., 2022), were applied throughout the training process. Furthermore, we utilized Automatic Mixed Precision (Micikevicius et al., 2018) in conjunction with 4 A6000 GPUs.

For the multimodal datasets, we customized two hyperparameter settings configuration based on the datasets: IEMOCAP dataset for fine-tuning and AVMNIST dataset for training from scratch. In the fine-tuning approach, particularly for the cases of [**L**→**A**] and [**A**→**L**] on IEMOCAP dataset, we conducted training for 30 epochs with a batch size of 4. Given that the learning procedure is fine-tuning, we adjusted the learning rate to $5 \times 10^{-5}$ and the weight decay to $5 \times 10^{-4}$, while also applying cosine learning rate scheduling (Loshchilov & Hutter, 2017). The Adam optimizer (Kingma & Ba, 2015) was employed, and no data augmentations were utilized. For training from scratch, specifically in the cases of [**L**→**V**] and [**A**→**V**] on the AVMNIST dataset, we again trained for 30 epochs with a batch size of 32. The learning rate was set to $1 \times 10^{-3}$, with a reduction factor of 0.1 applied at 25 epochs. The Adam optimizer (Kingma & Ba, 2015) was used, and data augmentations were omitted during this training phase as well.

## C  ADDITIONAL CASE STUDIES

### C.1  DESIGN CHOICES IN LATENT LOSS FUNCTION $\mathcal{L}_z$

In this section, we conduct an ablation study to assess the impact of different latent loss functions. Our primary approach employs mean squared error (MSE) loss to directly align latent representation vectors, we also explore the use of cosine embedding loss: $\mathcal{L}_z = \mathbb{E}\left[\mathbb{1} - \cos\left(g(\mathbf{x}_m^i), \hat{z}_m^j\right)\right]$. For comparison, we consider the best results of each cases from Table 1 and Table 2. As summarized in Table 6, the performance remains consistent across both loss functions, with variations ranging from $-0.1\%$ to $+0.7\%$. These results suggest that adhering to the representation distribution of the $M_j$ modality effectively supports training of $M_i$ modality, regardless of the choice of loss function.

Table 6: Performance comparison of different $\mathcal{L}_z$ across various datasets and cases

| Datasets & Cases | MSE | Cosine Embedding Loss |
|---|---|---|
| ImageNet-1K [**L**→**V**] | 81.90 | 81.97 |
| IEMOCAP  [**L**→**A**] | 61.20 | 61.20 |
| IEMOCAP  [**A**→**L**] | 56.49 | 56.77 |
| AVMNIST  [**V**→**A**] | 42.44 | 42.67 |
| AVMNIST  [**A**→**V**] | 66.69 | 66.25 |

### C.2  ADDITIONAL RESULTS IN THE LARGER MODEL

We also conducted experiments with a larger model, focusing on the [**L**→**V**] case on the ImageNet-1k dataset. Specifically, we utilized ViT-L/16, which requires approximately $5\times$ FLOPs compared to ViT-B/16. As the results are proved on Table 7, it demonstrate that our approach achieves significant improvements even with this larger model, where it shows similar gap compare to Table 1. This indicates that our method does not rely on the scalability of model size; instead, it leverages the qualitative representations of the $M_j$ modality to effectively synergize with or enhance the training of the $M_i$ modality.

Table 7: Large model classification results on ImageNet-1K and evaluation benchmarks

| Model [**L**→**V**] | IN | V2 | Rend. | Sketch | A | Style. | C ($\downarrow$) |
|---|---|---|---|---|---|---|---|
| ViT-L/16 (reproduced) | 80.04 | 70.75 | 51.33 | 39.73 | 31.01 | 25.63 | 44.85 |
| + RoBERTa (Liu et al., 2019) | 81.63 | 71.76 | 53.00 | 40.87 | 32.33 | 28.50 | 41.61 |

### C.3  DESIGN CHOICES FOR IMPERFECT SUPERVISION $\hat{z}^j$

In this section, we present additional experiments for the [**L**→**V**] case in ImageNet-1K classification to evaluate how varying cases of imperfect supervision impact performance. We define three different cases of imperfection setting to analyze its effects:

- **Case 1 - Completely Imperfect Supervision:** At this case, we generated 1,000 random sentences using ChatGPT-4o and randomly matched them to the training data of the $M_i$ modality model. This represents a highly imperfect supervision setting, introducing significant noise into the supervision of the $M_i$ modality.

- **Case 2 (Recap) - `"This is about Class #."`:** This case corresponds to our main approach, where supervision is based on random number `#` assigned to the data.

- **Case 3 (Recap) - Perfect Supervision:** As described in Section 5.2 and Appendix B.1, this case employs high-quality supervision generated by LLaVA for each image in the dataset.

The results in Table 8 reveal that even when $\hat{z}^j$ is entirely unrelated, representing a completely imperfect paired dataset, it can still synergize effectively with the $M_i$ modality. Notably, this setup achieves an improvement of nearly $1.5\%$ compared to training with a single modality and **Case 1** supervision. These findings underscore the strength of our approach and strongly align with our theoretical perspectives, as in Theorem 2.

Table 8: ImageNet-1K classification results with ViT-B/32 + RoBERTa under different cases of $\hat{z}^j$

| Model [L→V] | Single$^\dagger$ | Case 1 | Case 2 | Case 3 |
|---|---|---|---|---|
| Accuracy | 75.04 | 75.52 | 76.75 | 76.99 |

$^\dagger$: No RoBERTa, ViT-B/32 only.

### C.4 Additional Ablation Approaches constructing $\hat{\mathcal{Z}}_k$

Here, we conducted ablation studies to construct $\hat{\mathcal{Z}}_k$ using alternative multimodal learning approaches. While our method leverages $\mathcal{L}_z$ to enhance representation learning from the $M_j$ modality, we also evaluated traditional strategies such as concatenation and addition. For concatenation (late fusion), we followed the standard late fusion approach (Wang et al., 2020), where the representations from both modalities are concatenated into a single vector for joint training. Before concatenating, we applied an interpolation term $\alpha$, combining $(1 - \alpha) \cdot z^k$ and $\alpha \cdot \hat{z}^j$. For addition, we directly combined the interpolated representations with the coefficient $\alpha$ by adding them.

As shown in Table 9, traditional approaches result in significant performance degradation compared to our method, which utilizes the latent loss function $\mathcal{L}_z$. Similarly, Table 10 further demonstrates how our approach uses $\alpha$ to interpolate the representation space $\hat{\mathcal{Z}}^k$ between $\mathcal{Z}^i$ and $\hat{\mathcal{Z}}^j$, effectively reducing the latent representation gap. In contrast, concatenation and addition primarily introduce bias from the $M_j$ modality into the $M_i$ modality without effectively reducing the latent representation gap, where there are no additional loss term that reduces its gap. As the result, these conventional methods fail to construct an optimal $\hat{\mathcal{Z}}^k$, which, according to our theoretical hypothesis, should be embedded in the interpolated space between $\mathcal{Z}^i$ and $\hat{\mathcal{Z}}^j$. Therefore, these findings underscore the limitations of traditional multimodal learning approaches in our framework.

Table 9: Performance comparison of traditional multimodal learning approaches and applying $\mathcal{L}_z$

| Datasets & Cases | $\mathcal{L}_z$ | Concatenation | Addition |
|---|---|---|---|
| IEMOCAP [L→A] | 61.20 | 60.68 | 59.68 |
| IEMOCAP [A→L] | 56.49 | 55.90 | 55.34 |
| AVMNIST [V→A] | 42.44 | 42.07 | 41.49 |
| AVMNIST [A→V] | 66.69 | 65.44 | 65.59 |

Table 10: Gap (MSE Loss) between $\hat{z}^k$ and $\hat{z}^j$ across different approaches

| Datasets & Cases | $\mathcal{L}_z$ | Concatenation | Addition |
|---|---|---|---|
| IEMOCAP [L→A] | 0.129 | 0.892 | 0.906 |
| IEMOCAP [A→L] | 0.270 | 0.516 | 0.762 |
| AVMNIST [V→A] | 0.017 | 0.453 | 0.774 |
| AVMNIST [A→V] | 0.033 | 0.481 | 0.360 |

