# OpenReview forum: "Can One Modality Model Synergize Training of Other Modality Models?"
_ICLR.cc/2025/Conference — ICLR 2025 Poster_

### Official Review · Reviewer_Mtx8 · 2024-10-22

**Soundness:** 3
**Presentation:** 4
**Contribution:** 2
**Rating:** 5
**Confidence:** 4

**Summary:**

The paper demonstrates that performance on a target modality can be improved by leveraging another modality, even without paired samples. Both theoretical and empirical evidence are provided to support this claim, showcasing the method's effectiveness.

**Strengths:**

* The paper tackles an important problem: obtaining paired data is often challenging in many setups.
* The approach is backed by both theoretical analysis and empirical results, showing clear improvements over baseline methods.
* The paper is well-structured, with illustrations that help clarify key messages.

**Weaknesses:**

* The main contribution—showing that using an unpaired modality can improve performance—has already been explored in prior works. For instance, some studies demonstrate leveraging unpaired modalities through pretraining on one modality and fine-tuning on another, like from image to video to audio [1], or from text to image-text [2]. Additionally, the issue of handling unpaired or missing modalities has been addressed before, yet the paper does not discuss relevant works in this domain [3,4,5]. Including this discussion would better position the paper.

* It is not clear why the author decides to leverage other modality through an l2 loss between the features spaces. Other design choices can  be explored. For example, concatenation, addition or other multimodal features fusion techniques.

* The experiments use relatively small models on classification tasks. It remains unclear whether the proposed method would be effective on larger, more complex, maybe generative models (e.g., Multimodal LLMs, CLIP).

[1] Shukor, Mustafa, et al. "Unified model for image, video, audio and language tasks." TMLR (2023).

[2] Liu, Haotian, et al. "Improved baselines with visual instruction tuning." CVPR 2024.

[3] Kim, Donggeun, and Taesup Kim. "Missing Modality Prediction for Unpaired Multimodal Learning via Joint Embedding of Unimodal Models." ECCV (2024).

[4] Lee, Yi-Lun, et al. "Multimodal prompting with missing modalities for visual recognition." CVPR. 2023.

[5] Wang, Hu, et al. "Multi-modal learning with missing modality via shared-specific feature modelling." CVPR. 2023.

**Questions:**

Please check the weaknesses section, in particular the question about how leveraging other modalities is done.

---

> ### Author Response · Authors · 2024-11-22
>
> Dear reviewer, we deeply appreciate the reviewer's thoughtful feedback and constructive recommendations. Let us first address the concerns and insightful weaknesses & questions you raised.
>
> **Weaknesses & Questions:**
>
> **Weakness & Question 1: Comparisons to the suggested prior works**
> - While prior works, such as UniVal [1] and LLaVA [2] as suggested by the reviewer, have made significant advancements in leveraging multimodal learning with unpaired datasets, these contributions primarily rely on extensive empirical validations, not showing the theoretical analysis. Also, each work focuses on particular pairs of modalities. Our work emphasizes the theoretical answer to the fundamental question:  "How can multimodal datasets synergize with one another?" even when paired datasets are imperfect. Our theory works are not limited to particular modalities (we do not assume a certain modality in our theory) and specific methods to make the pair imperfect (we use the general notations of the imperfect paired modality $\hat{z}^j$ in our theory).
> - We hope to emphasize that our theory can be a fundamental tool for understanding how the existing works, including your suggested prior works, work well under unpaired supervision.
> - We appreciate the reviewer’s valuable observation regarding the insufficient discussion of prior research on unpaired or missing modalities, such as [3], [4], and [5], in our related works section. In the revised manuscript, we ensure that these studies are thoroughly incorporated and discussed in the revised version of the paper. We have updated our paper accordingly and appreciate the reviewer for bringing this to our attention.
>
> **Weakness & Question 2: Ablations on the loss term**
> - Our choice of L2 loss is intended to directly minimize the difference between the modality's representation space, ensuring semantic consistency. That said, we fully acknowledge the reviewers' suggestion to explore alternative latent loss functions, which could provide additional insights into the effectiveness of different approaches.
> - To address this, we have additionally tested the case with cosine embedding loss as the latent loss function, i.e., measuring the distance between two features from different modalities via cosine distance. As presented below, the results demonstrate the comparative performance across these methods, highlighting the minimal sensitivity for the form of loss term. We hope this addition provides a clearer understanding of the ablation studies on the latent loss term.
> - We want to emphasize that our work focuses on the theoretical understanding of the synergy between two modalities. Thus, we aim to use a simple yet direct loss term to align two modalities.
>
>     | **Datasets**  | **MSE (Paper)** | **Cosine Embedding Loss** |
>     | --- | --- | --- |
>     | ImageNet [L →  V] | 79.58 | 79.61 (+0.03%) |
>     | IEMOCAP [L →  A] | 61.20 | 61.20 (-0.00%) |
>     | IEMOCAP [A →  L] | 56.49 | 56.77  (+0.28%) |
>     | AVMNIST [V →  A] | 42.44 | 42.67 (+0.23%) |
>     | AVMNIST [A →  V] | 66.69 | 66.25 (-0.44%) |

---

> ### Author Response · Authors · 2024-11-22
>
> (continue)
>
> **Weakness & Question 3: Additional attempts with larger architectures**
>
> - Given the limited computing resources of our group, we have tried to conduct experiments using architectures as large as possible by adopting a Vision Transformer (ViT-B/16, ViT-B/32) from scratch, paired with RoBERTa-large. Compared with CLIP, suggested by the reviewer, we believe that the ViT versions used in our paper are sufficiently complicated architectures.
> - As shown in the following table, the comparison of FLOPs and throughput for each ViT model demonstrates that these models are comparable in scale to CLIP.
>
> |  | **ViT-B/32 + RoBERTa** | **ViT-B/16 + RoBERTa** | **CLIP-B/32** | **CLIP-B/16** |
> | --- | --- | --- | --- | --- |
> | FLOPs | $1.2\times10^9$ | $2.6 \times10^9$ | $0.7\times10^9$ | $2.0\times10^9$ |
> | img / sec | $2708.39$ | $1539.53$ | $3301.29$ | $1911.55$ |
> - However, we acknowledge the importance of addressing scalability concerns. Therefore, we have tried test with larger model ViT-L/16 + RoBERTa, which represents the upper limit of our lab-scale computing resources. This configuration has approximately 5 times the FLOPs of the largest model used in our original paper, ViT-B/16+RoBERTa. The results of training ViT-L/16 and ViT-L/16 + RoBERTa are provided below:
>
> |  | **IN** | **V2** | **Rend.** | **Sketch** | **A** | **Style** | **C ($\downarrow$)** |
> | --- | --- | --- | --- | --- | --- | --- | --- |
> | ViT-L/16 | 78.31 | 68.99 | 49.03 | 37.88 | 29.14 | 23.25 | 46.39 |
> | ViT-L/16 + RoBERTa | 80.71 | 70.62 | 52.50 | 40.45 | 31.85 | 27.47 | 41.61 |
>
> - The key aspect of our approach is its focus on addressing the fundamental question from a theoretical perspective. Consequently, our approach shows improvements in all results, even in larger model. demonstrates improvements across all results, even when using larger models. This outcome indicates that our approach is not reliant on model scalability. We have updated the manuscript at the appendix C to include these additional experiments accordingly.
> - Moreover, larger language models (LLMs) such as LLaMA, which are typically decoder-only models, differ in their fundamental design from our approach. Decoder-only LLMs are primarily optimized for generation tasks, whereas our work emphasizes obtaining high-quality representations from the language modality (e.g., the $M_j$ modality), rather than generative decoding process. Given this distinction, we believe the experimental results we present offer valuable insights and provide sufficient evidence to support our hypothesis.
> - Again, our work primarily focuses on the theoretical and practical aspects of multimodal learning rather than the scalability of individual models.
>
> \
> \
> Thank you for highlighting these thoughtful concerns. We hope our responses have provided the clarity you were seeking.

---

> ### Comment · Reviewer_Mtx8 · 2024-11-22
>
> Thanks for the additional experiments and clarifications. Given that the authors have addressed part of the concerns, I will not reduce my score. However, I find the authors response not convincing enough to increase my scores. For instance:
>
> * **Positioning with respect to other work:**  even if prior works addressing similar research problems rely on "extensive empirical validations" and "not showing the theoretical analysis", this is not a good reason to not discuss and position the paper with respect to these works.
>
> * **L2 loss:** I see the authors tested another loss to validate their work. However,  my suggestions was about doing architectural changes ("Other design choices can be explored. For example, concatenation, addition or other multimodal features fusion techniques"). Is  using an additional auxiliary loss  the only way to leverage other modalities? If this is not the case the reader should expect a convincing theoretical or empirical response.
>
> * **Scaling:** The current scale of the experiments does not validate the scalability to recent multimodal models. I see the authors points about focusing on the theoretical study and the lab constraints to scale more. But I think the scalability is still a concerns

---

> > ### Author Response · Authors · 2024-11-26
> > **To add further comments about L2 loss**
> >
> > First, thank you again for the thoughtful feedback and insights. We would like to provide additional comments regarding the latent loss function and its comparison to alternative design choices, particularly focusing on the L2 loss.
> >
> > **Further Comments on the L2 Loss**
> >
> > - We have included experimental results showing both accuracy and the gap (MSE) between $\hat{z}^k$ and $\hat{z}^j$ (refer to the updated tables). Here, we aim to analyze in more detail why alternative design choices, such as concatenation and addition, may not align well with our proposed approach.
> >
> > - Our approach is grounded in the theoretical principle of embedding the representation space into an interpolated space between the $M_i$ and $M_j$ modalities, achieved through the latent loss $\mathcal{L}_z$ with interpolation term $\alpha$. Unlike our method, concatenation and addition do not include any mechanism to explicitly reduce the gap between $\hat{z}^k$ and $\hat{z}^j$. In contrast, our method incorporates representations from both modalities directly into the loss function, which helps minimize this gap. Therefore, we believe that using L2 loss (or cosine embedding loss) is more suitable for our framework compared to concatenation or addition.
> >
> > - To address this point in detail, we have added the above analysis to Appendix C.4 in the revised manuscript
> >
> > \
> > We sincerely appreciate the opportunity to consider and reflect on alternative approaches. Thank you for your valuable suggestions, which have helped us strengthen our work.

---

> > ### Author Response · Authors · 2024-12-02
> > **A Kind Reminder [Deadline Approaching]**
> >
> > **Dear Reviewer Mtx8,**
> >
> > It has been a privilege to refine our paper based on your insightful feedback. We sincerely hope that our previous responses, particularly regarding the design of loss functions, have addressed your concerns, if not entirely, then at least in part. As the deadline for your feedback approaches in approximately one day ```(deadline Dec. 2nd)```, and with two days remaining for us to submit our responses ```(deadline Dec. 3rd)```, we kindly offer this reminder of the timeline. If you have any additional feedback or suggestions, we would be truly grateful to receive them within given period.
> >
> > Thank you once again for your time and dedication in reviewing our work.

---

> ### Author Response · Authors · 2024-11-24
>
> We are truly grateful for your insightful recommendations, which we feel will considerably strengthen the impact of our paper. We will address each point below.
>
> \
> **Positioning with respect to other work:**
>
> - First, we sincerely appreciate your thoughtful insights and apologize for our earlier response regarding the related works section. Our intention was to emphasize the theoretical aspects and highlight subtle differences from prior works. However, we acknowledge that we did not sufficiently discuss the prior works, and we take full responsibility for this oversight.
> - To clarify the difference between our approach and prior works, most existing methods focus on training or fine-tuning both modalities simultaneously [1,2], training separate classifiers for each modality [3], training prompts [4], or using a shared encoder [5] to handle missing modalities . In contrast, our method leverages the latent representations from the $M_j$ modality (with frozen parameters) and synergize with the representations of the $M_i$ modality (training from scratch), rather than jointly optimizing both modalities. This means that no additional training is conducted on the $M_j$ modality; it is only used to extract its representation features.
> - We have added this discussion of related works in revised manuscript.
>
> **Ablations of latent loss function**
>
> - We also appreciate your valuable suggestion to include additional ablations. Following the reviewer’s recommendation, we conducted further experiments to compare the performance of different loss functions, including **MSE (as used in our paper)**, **Cosine Embedding Loss**, **Concatenation**, and **Addition**, [6,7] which are commonly used in multimodal learning. The results are summarized below:
>
>     | Accuaracy | **MSE (Paper)** | **Cosine Embedding Loss** | **Concatenation** | **Addition** |
>     | --- | --- | --- | --- | --- |
>     | IEMOCAP [L →  A] | 61.20 | 61.20  | 60.68 | 59.68 |
>     | IEMOCAP [A →  L] | 56.49 | 56.77   | 55.90 | 55.34 |
>     | AVMNIST [V →  A] | 42.44 | 42.67 | 42.07 | 41.49 |
>     | AVMNIST [A →  V] | 66.69 | 66.25  | 65.44  | 65.59 |
> - The results indicate that the **Concatenation** and **Addition** approaches do not show significant improvements compared to either single-modality learning or our proposed method. To further analyze this, we conducted additional experiments comparing the MSE loss for our approach against the Concatenation and Addition methods, as shown in the table below.
>
>     | Latent Loss | **MSE (Paper)** | **Concatenation** | **Addition** |
>     | --- | --- | --- | --- |
>     | IEMOCAP [L →  A] | 0.129 | 0.892 | 0.906 |
>     | IEMOCAP [A →  L] | 0.270 | 0.516 | 0.762 |
>     | AVMNIST [V →  A] | 0.017 | 0.453 | 0.774 |
>     | AVMNIST [A →  V] | 0.033 | 0.481 | 0.360 |
> - Based on these results, we hypothesize that the limitation of the **Concatenation** and **Addition** approaches arises from the convergence behavior of the latent loss. Our method directly minimizes the gap between the latent representations of each modality through targeted optimization. In contrast, **Concatenation** and **Addition** primarily transfer a bias from the $M_j$ modality's latent representation into the $M_i$ modality, without effectively reducing or addressing the latent gap between them. As the result, the overall approach of the **Concatenation** and **Addition** approaches is not well-suited to our proposed framework.
> - We hope this clarifies our choice of the latent loss.
>
> **Scaling Issue**
>
> - We sincerely appreciate the reviewer pointing out the scalability issue. We truly apologize for not being able to present additional results now on larger models due to computational resource constraints. This remains an empirical challenge in our current situation and has been left as future work. However, we believe that our theory is not dependent on the scalability issue and may still apply to larger models.
> - To clearly mention the limitations in scaling, we have added at **Limitation** about this issue in the revised manuscript.
>
> [1] Shukor, Mustafa, et al. "Unified model for image, video, audio and language tasks." TMLR (2023).
>
> [2] Liu, Haotian, et al. "Improved baselines with visual instruction tuning." CVPR 2024.
>
> [3] Kim, Donggeun, and Taesup Kim. "Missing Modality Prediction for Unpaired Multimodal Learning via Joint Embedding of Unimodal Models." ECCV (2024).
>
> [4] Lee, Yi-Lun, et al. "Multimodal prompting with missing modalities for visual recognition." CVPR. 2023.
>
> [5] Wang, Hu, et al. "Multi-modal learning with missing modality via shared-specific feature modelling." CVPR. 2023.
>
> [6] Tsai, Yao-Hung Hubert, et al. "Learning factorized multimodal representations." ICLR (2019)
>
> [7] Wang, Weiyao, Du Tran, and Matt Feiszli. "What makes training multi-modal classification networks hard?." CVPR. 2020.
>
> \
> Again, thank you for your valuable and intuitive insights.

---

### Official Review · Reviewer_E3nE · 2024-10-29

**Soundness:** 3
**Presentation:** 4
**Contribution:** 3
**Rating:** 8
**Confidence:** 4

**Summary:**

This paper investigates whether one modality model can enhance the training of another modality model without requiring paired multimodal supervision. The authors propose both theoretical and empirical evidence that imperfect supervision from one modality (e.g., language) can improve the performance of another modality (e.g., vision). They establish mathematical foundations showing that an interpolated representation between two modalities can outperform single-modality representations, even with imperfect cross-modal supervision. The work is validated through extensive experiments across vision, language and audio modalities, demonstrating consistent performance improvements. For example, in the vision domain, they show improvements of 1.5-2.5% on ImageNet classification and similar gains on robustness benchmarks by leveraging simple language prompts during training.

**Strengths:**

- Novel insights: The work challenges common assumptions about requiring paired supervision for multimodal learning and demonstrates unexpected improvements using single-modality pre-training and synergies between seemingly unrelated modalities.

- Strong theoretical foundation: The paper provides rigorous mathematical proofs for how and why cross-modal learning can work without paired supervision, establishing bounds on the interpolation coefficient $\alpha$ and showing the existence of superior interpolated representations.

- Comprehensive empirical validation: The authors demonstrate their approach across multiple modality pairs (Vision-Language, Vision-Audio, Language-Audio) and various architectures, showing consistent improvements across different settings and tasks. Results include not just standard classification metrics but also out-of-distribution generalization and robustness benchmarks, showing broad improvements across different evaluation criteria.

**Weaknesses:**

- Other language supervision: The language prompts used (e.g., "This is about Class #") are quite basic. It would be interesting to see how the method performs with more complex or varied language supervision.

- Theoretical assumptions: Some theoretical assumptions (e.g., Assumption 1 about $\Delta_{ij} \ge  0$) could benefit from more empirical validation or discussion of when they might not hold.

**Questions:**

- Figure 2 right, $Z_i$ should be $Z_j$.

---

> ### Author Response · Authors · 2024-11-22
>
> Dear reviewer, we extend our sincere gratitude to the reviewer for their insightful comments and valuable suggestions. We would like to take this opportunity to address the concerns and weaknesses you raised.
>
> **Weaknesses**
>
> **Weakness 1: More complex language supervision**
> - As demonstrated in Table 4 and Appendix B, we utilized captions generated by LLaVA and applied them to our method, resulting in more "perfect" representations compared to the simplified language prompts used in our initial experiments (see the column with $z^j$ without ‘hat’). While this refinement led to a slight improvement in performance, the gain was not substantial. This finding highlights a key contribution of our work: even with imperfect supervision for $M_j$ modality, it remains possible to effectively synergize and enhance the training of models for the $M_i$ modality.
> - By doing so, we tested the two extreme cases: the basic supervision ("This is about Class #,” used in our experiments) and the rich supervision (from LLaVA), and they showed a minimal gap. This implies that a case between “basic” and “LLaVA,” i.e., moderately complicated supervision, would show a similar performance with a minimal difference.
>
> **Weakness 2: Consideration of Assumption 1**
> - We assumed that $\Delta_{ij}$ might not converge to 0 because models trained individually on different modalities may not fully share their representation spaces. For example, even in CLIP, which aims to align two representations from different modalities, the representation spaces for different modalities are not identical [1].
> - Furthermore, as illustrated in Figure 3, we empirically visualize and compute the Wasserstein Distance, demonstrating how distinct these spaces remain.
>
> [1] Interpreting and Analyzing CLIP’s Zero-Shot Image Classification via Mutual Knowledge
>
>
> **Questions**
> - We have changed the notations on Figure 2, $Z_i \rightarrow  Z_j$.
>
> \
> \
>  We hope our responses have addressed your concerns and provided the necessary clarifications.

---

> > ### Comment · Reviewer_E3nE · 2024-11-25
> >
> > Thanks for your response. The rebuttal addressed my concerns and I will maintain my positive evaluation.

---

> > > ### Author Response · Authors · 2024-11-26
> > >
> > > Dear reviewer, we sincerely thank you for your insightful suggestions and thoughtful concerns. Your feedback has been invaluable in helping us improve weaknesses and strengthen our paper. We appreiciate for the time and expertise you have invested in reviewing our work.
> > >
> > > Thank you for your valuable contribution to the peer review process.

---

### Official Review · Reviewer_hzXr · 2024-11-01

**Soundness:** 3
**Presentation:** 3
**Contribution:** 3
**Rating:** 6
**Confidence:** 3

**Summary:**

The paper aims to answer whether imperfectly aligned paired data from other modalities can help learning in a multimodal setting.
Specifically, the authors propose an additional latent loss, to directly align the target modalities' latent representation with that of the output of a pre-trained encoder of the secondary (supportive) modality. The authors introduce and study a theoretical framework and show that even imperfect paired data can help approximate a hypothetical, perfectly aligned representation. They further demonstrate empirically that the additional latent loss led to stronger performance of the target modalities' encoder across various tasks and modalities.

**Strengths:**

The work considers a range of relevant modalities and conducts experiments across language, vision, and audio in various cross-combinations.
The mathematical framework introduced is intuitive and easy to follow.

**Weaknesses:**

The paper asserts in multiple locations that prevailing multimodal learning methods require "perfectly paired datasets" (quote from the introduction) between modalities. This, in my opinion, is not accurate an accurate representation of the thinking in the field. The authors cite CLIP as a notable multimodal model, which famously is trained on noisy web-scale paired data in the form of image alt text. Training multimodal models on noisy labels such as alt-text from web-scraped images is common practice and widely established (Radford et al. 2021, Dosovitskiy et al. 2020.,...). While improving the alignment of the training modalities is generally seen as desirable (e.g. Fang et al., 2023), "perfection" seems not a requirement.
The paper does not cite and / or discuss other works in the space of aligning multiple modalities without direct paired supervision data, including popular works such as ImageBind (Girdhar, et al., 2023) or 4M (Mizrahi et al., 2024). These works do not (solely) rely on paired multimodal data, seemingly directly addressing the limitations discussed by this work in section 2.3.
This, together with arguably understating how much alignment on noisily paired data has been previously studied in the field, arguably limits the novelty of this work.

One of the main statements of this work is that the label does not need to be perfectly paired / can be noisy. However, the transformations to introduce this noise studied in the work may not be sufficiently realistic. For example for the L -> V case, the noisy label \hat z^j is constructed by embedding the text "This is about (Class|Emotion) #." as per table 5. In terms of the measured downstream task, which is classification, this label is arguably not noisy, but perfectly represents the target task. In table 4 it is shown that changing this supervision signal with a caption produced by LLaVA leads to only minor improvement, which may not be surprising in this setting. (See the Questions section for suggestions around this.)

**Questions:**

Regarding Remark 2.1. "δ does not hinder the synergy", you say that
"We can extract an imperfect feature representation from Pj by giving
imperfect input to the modality Mj . This allows ˆzj exist in the distribution Pj 2. Consequently, ˆzj
is closer to or part of the latent space of the Mj than to that of the Mi or the true latent space."

Wouldn't this quite directly depend on how far \hat P_j is from P_j, as well as how aligned P_i is to P_j to begin with? Surely one could construct counter examples with adversarially chosen δ? This may not be much of a practical concern for reasonably close \hat P_j but is this statement not a bit strong in the general case?
Expanding on this, perhaps this could be empirically verified by exploring different levels of noise to introduce in \hat z^j, particularly in the L -> V task as suggested above. Have you perhaps already considered / explored different noising functions and compared their impact?

For the V -> A case in AVMNIST, you say "For the [V→A] case with AVMNIST, we use randomly shuffled images from AVMNIST as ˆzj in audio classification tasks". Could you clarify the random sampling in this case? Is it a random image from the entire dataset or a random image from the samples of the same target class? If it is a random image (i.e. unrelated to the paired modality at all), this seems significantly more "noise" than in other settings, it'd be great to understand a bit better to understand the motivation for this choice and perhaps similar experiments for other modalities.

On a general level, if we assume that the target distribution for a modality encoder g_i is similar to the one of a pre-trained encoder g_j of a different modality, the proposed latent alignment loss has some similarity to knowledge distillation. In this field there's been some notable prior work that suggests that the success of KD is partially attributable not only to a superior knowledge of the teacher but also to benefits of the training strategy itself. Notably, Born Again Networks (Furlanello et al., 2018) suggests a simple strategy of self-distillation can improve performance. Yuan Li et a., 2020 explores this further in "Revisiting Knowledge Distillation via Label Smoothing Regularization". This is relevant to this work since it could suggest a different mechanism leading to the empirically observed improvements that is less about multimodal transfer and perhaps more about a sort of regularization effect of the added latent loss. Has this been considered?

---

> ### Author Response · Authors · 2024-11-22
>
> Dear reviewer, we sincerely thank for the  thoughtful perspectives and insights. We would like to begin by addressing the concerns you raised regarding the weaknesses in our work
>
> **Weaknesses**
>
> **Weakness 1: About “perfect vs. imperfect” paired modalities**
> - First, we acknowledge the opinion that the existing multimodal training is basically with a “noisy” paired label, e.g., CLIP with web-scraped paired samples. However, we want to point out that the prior multimodal approaches rely on pairs of modalities that provide each other with detailed or simplified semantics. For example, consider an image of a poodle sleeping on a sofa. Then, for instance, the web-scraped textual descriptions can be “The poodle is on a sofa” as a detailed description, “The poodle is sleeping,” or even “The dog is sleeping” as a simplified description. While differing in detail, all these descriptions convey the essential semantic content—that a dog or poodle is present—preserving the primary object and concept in the image. That is why we referred to such descriptions as "perfect" because they retain critical semantic elements relevant to the main object.
>  - Conversely, we used "imperfect" to denote cases where descriptions lack meaningful semantic information about the primary object. For instance, instead of identifying a "dog," a description might label it as "class 1," "class 2," or "class N." These labels, while potentially relevant in specific contexts, do not inherently convey any information about a dog. In this sense, "imperfect" descriptions provide minimal semantic richness. For comparison, "noisy supervision" (e.g., LAION-400M) still contains more contextual information than what we define as "imperfect" descriptions.
> - Also, as you pointed out, in the [L→V] experiments, "This is about Class #." probably provides the information for visual tasks because the different index (#) can be a clue for discriminating different objects for visual classification tasks. To address the reviewer's concern directly, we expanded our experiments to include a more complex setting for comparison. Specifically, we imposed textual descriptions such as "This is about Class #," where "#" is randomly assigned to fixed classes across iterations. For instance, the description of a dog could be "Class 1" in the first iteration and "Class 1423" in the next.  Also, # can be chosen from an arbitrarily wider range of integers than the number of the existing image categories for the downstream visual tasks, which means that the textual description does NOT provide information about the visual semantics or tasks. The results of these experiments are summarized as follows.
> | **ImageNet 1K [L → V]** |   Original **$\hat{z}^j$**      | **Revised $\hat{z}^j$** |
> | ----------------------- | ----------------------- | ----------------------- |
> | ResNet-50 + RoBERTa         | 76.77                   |  76.90 (+0.13%)      |
> | ViT-B/32 + RoBERTa             | 74.97                   | 74.92 (-0.05%)          |
> | ViT-B/16 + RoBERTa             | 79.54                   | 79.58 (+0.04%)          |
>
> - The performance changes are minimal despite the coarse (imperfect in our notations) paired texts. The results appear consistent with our key findings, emphasizing, “Even an imperfect supervision can synergize the other modalities.”
> - We think it is hard to say that our paired supervision is “perfect” or “not noisy.” Also, the minimal gaps in Table 4 do not imply the perfection of our pairing but emphasize that imperfect supervision is sufficient to promote the training of other modalities.
>  - We hope this clarifies our intended use of these terms and provides additional context to address any concerns. Additionally, we have revised the related experiments for [L→V] cases in the revised manuscript. We will revise the additional experiments (e.g., ViT-B/32 + BERT, ViT-B/16-BERT, ResNet-50 + BERT) on Table 1 as soon as the results become available.
> - We appreciate your thoughtful review and hope to hear your feedback during the discussion period.
>
> **Weakness 2: About the missing related works and our emphasis on theory**
> - We apologize for not citing prominent works like ImageBind or the 4M paper, which demonstrated impressive performance without relying on directly paired datasets. Our emphasis beyond these works is theoretically answering the foundation question: "**How can one modality model effectively promote training the other modality across diverse settings, even with 'noisy' or 'imperfect' supervision?**" Our theory can explain how the existing multimodal methods, including ImageBind and 4M, and our experiments work well. While many recent works have achieved surprising empirical results in multimodal learning, relatively few have investigated its theoretical viewpoint. Therefore, we aim to address this gap.
> - We have incorporated the suggested works into our revised paper and strengthened our theoretical novelty to provide a more comprehensive view.

---

> > ### Comment · Reviewer_hzXr · 2024-11-22
> >
> > I want to sincerely thank the authors for their well considered rebuttal. I particularly appreciate the additional experimental results provided.
> > I will address the individual points below.
> >
> > Regarding the level of noise in a “noisy” paired label in e.g., CLIP, I understand the distinction the authors are making. To the examples provided in the rebuttal I’d add that in practice many examples of alt text may also have little to no relation to the semantic content of the image. For example instead of “The dog is sleeping” the alt text may be “(c) John Doe” or “thumbnail”. Filtering such data to achieve better alignment is a fairly active topic in the field, for example discussed in [1]. Because of this, I personally would still not describe such data as perfectly paired.  Still, I believe the difference you aim to stress is that in your work you focus on the mechanisms how even without [any] well aligned data points one can still achieve improvements and that surely is still a different goal.
> >
> > Regarding the additional setting discussed in your rebuttal where you randomize the class ID per iteration, first let me thank you again for adding this, I believe this is directly addresses my concern. I see that in the revised manuscript you specifically mention how the label is now entirely unrelated to the ground truth class label (L374) which I believe significantly strengthens the results you present.
> >
> > Regarding the missing citations, thank you for adding them.
> >
> >
> > [1] Yu, Haichao, et al. "The devil is in the details: A deep dive into the rabbit hole of data filtering." arXiv preprint arXiv:2309.15954 (2023).

---

> ### Author Response · Authors · 2024-11-22
>
> Then, we would like to this discuss about the concerns and questions.
>
> **Question 1: Regarding Remark 2.1**
> - As an answering your questions: “this quite directly depend on how far $\hat P_j$ is from $P_j$, as well as how aligned $P_i$ is to $P_j$ to begin with”, we hypothesized that $\Delta_{ij}$ is greater than \delta. While it may seem counterintuitive, given that models like CLIP align representations across modalities to share semantics, our approach begins with distinct modalities without prior shared representations.
> - As mentioned in our previous response, we define $\hat{P_j}$ and $P_j$ as the distributions for modality $M_j$, with the hypothesis that $\Delta_{ij}$ is greater than \delta. Since $\hat{P_j}$ and $P_j$ represent "imperfect" and "perfect" supervision within modality $M_j$, both distributions still remain within the representation space of $M_j$. In contrast, $P_i$ represents a distinct modality, likely forming a very different representation space. Thus, we believe that our hypothesis—that $\Delta_{ij}$ is larger than $\delta$—represents a general case and is well-founded.
>
> **Question 2: Additional results with more noisy settings**
> - To clarify the “random sampling” in the case of [V → A], we shuffled within the entire vision dataset in AVMNIST to create an imperfect supervision setting.
> - We also acknowledge the reviewer's observation that this approach introduces more imperfection compared to other cases, such as [A → V] and [A → L]. Based on this feedback, we applied similar settings to the other cases.
> - For instance, in the [A → V] and [A → L] cases, we introduced Gaussian noise to each audio data as described in our original version of the paper. Then, we additionally introduced random shuffling across the entire audio dataset to increase imperfection in supervision. Additionally, for [L → A] case, we followed a similar process for generating $\hat{z}^j$ as implemented in the [L → V] case.
> - The results, summarized in the table below, show very minimal differences compared to the findings reported in the original paper:
>
>     | Datasets | **Original $\hat{z}^j$** | **Revised $\hat{z}^j$** |
>     | --- | --- | --- |
>     | IEMOCAP [L →  A] | 61.29 | 61.20 (-0.09%) |
>     | IEMOCAP [A →  L] | 56.45 | 56.49 (+0.04%) |
>     | AVMNIST [V →  A] | 42.44 | 42.44 |
>     | AVMNIST [A →  V] | 66.56 | 66.69 (+0.13 %) |
>
>     These findings suggest that our initial approach effectively captures the intended level of imperfection. We have revised the manuscript accordingly to include these additional experiments. (Table 2 & Table 3)
>
> **Question 3: Prior approaches using knowledge distillation**
> - We appreciate the reviewer's observation regarding the similarity of our approach to Knowledge Distillation (KD), where a model for modality $M_j$ teaches a model for modality $M_i$. However, our method differs significantly from traditional KD approaches, such as self-distillation (Furlanello et al., 2018) or De-KD (Yuan Li et al., 2020).
> - First, KD typically focuses on transferring knowledge through the distillation of classifier outputs, where final knowledge is derived from label supervision. In contrast, our approach emphasizes learning the representation distribution of modality $M_j$ while simultaneously training the representation space of modality $M_i$. As highlighted in our theoretical framework, the core of our method lies in aligning the centers of the representation distributions between the two modalities. This approach ensures that the representations of both modalities, trained independently, are effectively bridged to synergize their latent spaces.
>
> \
> \
> We hope these clarification addresses the reviewer's overall concerns.

---

> > ### Comment · Reviewer_hzXr · 2024-11-22
> >
> > Thank you very much for your thoughtful responses.
> >
> > Regarding the response to question 1, I agree that particularly in the case you mention where M_i and M_j are distinct modalities without prior alignment, this assumption would almost surely hold in practice, as I also indicated in my original review. I was only suggesting that it may be possible to construct a case where this may not be true. If that possibility exists, I think it may be better to slightly soften the language in the manuscript to allow for that, which I think can be done without reducing the claims in practice. Concretely, I think this could be as simple as to change “[…] δ will be much smaller than both […]” to something like “[…] δ will generally be much smaller than both […]”, i.e. just add a “generally” to correspond to the “unlikely” in the previous part of that sentence. In my opinion this just allows for the possibility of outliers / adversarial cases without changing the overall statement.
> >
> > Regarding the response to question 2, again I want to thank the authors for the additional experiments conducted. Conducting such ablations under the timelines of a review period is surely not easy and the effort is deeply appreciated. Again, I believe the results significantly strengthen the contributions. Similar to the ablation with LLaVA captions you included in the paper, I’d even suggest that how little difference the choice of noising function itself has on results is in its own a bit noteworthy, complementing and strengthening the insights from table 4. Since you have all the experimental results already, it may be worthwhile to add them to the appendix? But just a suggestion.

---

> > > ### Author Response · Authors · 2024-11-24
> > >
> > > Dear Reviewer, we sincerely appreciate your worthwhile suggestions, which we believe will significantly strengthen our paper. We will address each points below.
> > >
> > >
> > >
> > > **Recommandation for word choices at Remark 2.1**
> > >
> > > - First, we are grateful for your recommendation to improve the wording of Remark 2.1. We will rephrase the word to use smoother term *generally* rather than overall cases.
> > > - We have included this term in the revised manuscript.
> > >
> > > **Design choices for imperfect supervision $\hat{z}^j$**
> > >
> > > - Additionally, we appreciate your proposal to include further ablations by varying the design of the imperfect supervision $\hat{z}^j$ at different levels. Following your suggestion, we conducted **additional experiments on ViT-B/32 + RoBERTa**, summarized as follows:
> > >     - Level 1: Random sentences (Completely imperfect supervision)
> > >     - Level 2: “This is about class #.’ # → random number. (Our and reviewer’s suggestion previously)
> > >     - Level 3: Related Supervision (Generated supervision via LLaVA)
> > >
> > >       |  | Single Modality | Level 1 | Level 2 | Level 3 |
> > >       | --- | --- | --- | --- | --- |
> > >       | Accuracy | 72.39 | 74.40 | 74.92 | 75.03 |
> > >
> > > - As the results, it still shows improvement  We hypothesize that this is because the $M_i$ modality model attempts to learn a comprehensive representation space from $M_j$ modality, consistent with our theoretical findings.
> > > - We have included these results in Appendix C.3 for revised manuscript.
> > >
> > > \
> > > Once again, we thank you for your valuable feedback and insightful suggestions.

---

> > > > ### Comment · Reviewer_hzXr · 2024-11-27
> > > >
> > > > I again thank the authors for all the improvements made. I have raised my score to 6, recommending acceptance.

---

> > > > > ### Author Response · Authors · 2024-11-27
> > > > >
> > > > > Dear reviewer, we greatly appreciate the time and effort you have dedicated to evaluating our work. Your thoughtful guidance has significantly improved our work, and we are sincerely grateful for your valuable contribution to our academic endeavor.
> > > > >
> > > > > \
> > > > > Thank you for your significant contribution to the peer review process.

---

### Author Response · Authors · 2024-11-22
**Overall Comments: Summary of revisions to the original paper**

Dear all reviewers, we truely appreciate your feedbacks. Here, we first provide what we revised on the original paper based on your thoughtful comments. (We have marked **all revised part in blue** for the clarity.)

- **For Reviewer hzXr:**
    - We have **updated** the results in **Table 1**.
    - We have revised **Table 2 and Table 3** to address your comments.
    - **Missing references to related works** on unpaired multimodal settings have been included in **Section 2.3**.
    - We have provided modified description on **Additional Settings $\textit{(how to get $\mathbf{\hat{z}^j}$)}$** to **clarify the imperfect and noisy supervision**.
   - We have added the word 'generally' to Remark 2.1 to improve the wording for smoother condition.
   - We have included additional experimental results on **design choices for imperfect supervision $\hat{z}^j$ in Appendix C.3.**
- **For Reviewer E3nE:**
    - We haved **changed notations** on **Figure 2**.
- **For Reviewer Mtx8:**
    - We have **updated missing related works** on unpaired multimodal settings on **Section 2.3**.
    - We have conducted additional experiments using **different loss terms**, as presented in **Table 6 in Appendix C.1**.
    - We also provide additional experiment results for **larger model configurations** in **Table 7 in Appendix C.2.**
    - We have **included further related works and filled in missing details in Section 2.3.**
    - The **scalability issue** has been addressed in the **Limitation section (Section 6: Further Discussion).**
    - We provided additional experiements results with analysis for other design choices for latent loss in Appendix C.4

---

### Meta-Review · Area_Chair_iHYq · 2024-12-20

**Metareview:**

This paper investigates whether one modality model can enhance the training of another modality model without requiring paired multimodal supervision. The authors propose both theoretical and empirical evidence that imperfect supervision from one modality (e.g., language) can improve the performance of another modality (e.g., vision). The analysis reveals that interpolated representations between two modalities can outperform single-modality representations, even with imperfect cross-modal supervision, and they empirically validate this work through experiments across vision, language and audio modalities with consistent improvements.

Reviewers were split with 1 marginal reject, 1 marginal accept, and 1 accept. They generally found the work interesting and useful, with strong results across language, vision, and audio in various cross-combinations, and that the theory is intuitive and easy to follow.

Key weaknesses pointed out by the reviewers include that more comparisons with prior work, that using an unpaired modality to improve performance had been explored, larger-scale experiments, and more ablation studies. From what I've seen, the authors have provided additional experiments on all of these concerns, and I am satisfied with them, so I recommend acceptance.

**Additional Comments On Reviewer Discussion:**

Reviewer hzXr raised from 5 to 6 after the discussion since they felt their main concerns had been addressed, which I agree. Most of these concerns were comparisons with prior work and clarifications on the weak pairings/noisy pairing settings. Reviewer E3nE maintained their score of 8.

Reviewer Mtx8 maintained their score of 5. Their main concerns were that the main contribution—showing that using an unpaired modality can improve performance—has already been explored in prior works, requesting more discussions wrt prior work, requests for ablations of loss functions beyond L2 loss to interpolate modality representations, more architectural ablations to fuse modalities, and increasing the scale of the models in experiments. From what I've seen, the authors have provided additional experiments on all of these concerns, and I am satisfied with them.

---

### Decision · Program_Chairs · 2025-01-22

Accept (Poster)